# Exonic enhancers are a widespread class of dual-function regulatory elements

Jean-Christophe Mouren [1], Magali Torres [1,2,3],
Antoinette van Ouwerkerk [1,2,3], Iris Manosalva [1,2], Frederic Gallardo [1],
Salvatore Spicuglia [1,2,4] & Benoit Ballester [1,4] ✉

Exonic enhancers (EEs) occupy an under-appreciated niche in gene regulation. By integrating transcription factor binding, chromatin accessibility, and high-throughput enhancer-reporter assays, we demonstrate that many protein-coding exons possess enhancer activity across species. These candidate EEs (cEEs) exhibit characteristic epigenomic signatures, form long-range interactions with gene promoters, and can be altered by both nonsynonymous and synonymous variants. CRISPR-mediated inactivation demonstrated the involvement of cEEs in the cis-regulation of host and distal gene expression. Through large-scale cancer genome analyses, we reveal that cEE mutations correlate with dysregulated target-gene expression and clinical outcomes, highlighting their potential relevance in disease. Evolutionary comparisons show that cEEs exhibit both strong sequence constraint and lineage-specific plasticity, suggesting that they serve ancient regulatory functions while also contributing to species divergence. Our findings expand the landscape of functional elements by establishing cEEs as a component of gene regulation, while revealing how coding regions can simultaneously fulfil both protein-coding and cis-regulatory roles.

Precise control of gene expression relies on a complex network of regulatory elements that integrate spatial, temporal, and cell-type-specific signals. Traditionally, these regulatory elements have been localised to noncoding regions of the genome, where enhancers serve as key modulators of transcriptional activity[1]. However, accumulating evidence points to additional regulatory roles within protein-coding sequences, challenging the assumption that coding regions merely encode proteins[2–11].

The concept of "exonic enhancers" (EEs) has arisen from observations that certain exons, despite encoding protein domains, also exhibit features comparable to enhancers, such as open chromatin, transcription factor (TF) occupancy, and histone modifications associated with active regulatory elements[2–11]. Historically, single-locus experiments already showed bona fide enhancer activity embedded in coding sequence, for example, a regulatory element within a coding exon of *KRT18*[2] and two ultraconserved *HOXA2*

exonic enhancers that drive hindbrain expression[3,5], the latter explicitly mapped via synonymous-constraint scans[6]. Subsequent comparative and functional studies expanded this view, showing that coding exons can act as tissue-specific enhancers[4,8,10] and highlighting clinical relevance[12]. Together with genome-wide TF occupancy in exons[7], these early reports established that enhancer activity in coding regions is a real and recurrent phenomenon rather than an anecdotal curiosity.

Building on these observations, our study provides the first systematic, cross-species census of exons that meet stringent enhancer criteria. Our analysis asks three quantitative questions that have not been addressed at genome scale: (i) How common are such dual-function exons? (ii) How potent is their enhancer activity? (iii) What selective and mutational constraints accompany the coexistence of coding and regulatory information? By integrating >20,000 TF-binding profiles, multi-omic enhancer marks, high-throughput

[1]Aix Marseille Univ, INSERM, TAGC, Marseille, France. [2]Equipe Labellisée LIGUE, Marseille, France. [3]These authors contributed equally: Magali Torres, Antoinette van Ouwerkerk. [4]These authors jointly supervised this work: Salvatore Spicuglia, Benoit Ballester. ✉e-mail: benoit.ballester@inserm.fr

reporter assays, and comparative genomics, we provide a panoramic view of candidate Exonic Enhancers (cEEs) across four phyla.

We confirm that many exons act as both coding and regulatory sequences, revealing an under-appreciated layer of genomic complexity that challenges the classic coding-non-coding distinction. Variants in these dual-purpose elements can affect both protein function and transcription, amplifying their phenotypic and disease impact; a perspective first advocated by Ahituv et. al[12] and now substantiated by our exon-only STARR-seq screens and CRISPRi perturbations. Conceptually, we (a) corroborate earlier locus-specific findings with orthogonal, exon-bounded assays, (b) extend them genome-wide and across species, and (c) introduce new analyses that relate exonic-enhancer activity to protein-structural context and variant sensitivity.

Here we present a comprehensive analysis of EEs across vertebrate and invertebrate genomes using large-scale integration of TF-binding maps, open chromatin assays, and high-throughput enhancer-reporter data. We systematically identify thousands of candidate EEs (cEEs), validate their enhancer-like properties, and explore their functional relevance in various biological contexts. By intersecting cEEs with cancer genomic datasets, we demonstrate that variants within these coding-regulatory regions could influence gene expression and patient outcomes, suggesting broad clinical implications. Finally, through multi-species comparative analyses, we reveal that cEEs display both deep evolutionary conservation and lineage-specific diversification, reflecting selective pressures to maintain dual coding-enhancer functions while enabling adaptive regulatory plasticity.

Collectively, our findings confirm EEs as an integral layer of gene regulation and underscore the need to look beyond non-coding DNA by recognising that coding exons can direct transcription. Appreciating the regulatory capacity of exons reframes how we interpret variants, dissect pathogenesis and conceive genome architecture, justifying assessment of coding mutations for regulatory effects[12].

## Results

### Exonic Enhancers in four species revealed by TF binding maps

To detect protein-coding exons harbouring regulatory activity, we integrated large-scale TF ChIP-seq profiles from ReMap[13] to construct a high-resolution map of TF occupancy in coding sequences. We observed a pronounced enrichment of multi-TF binding in certain internal exons, as illustrated by the *GARRE1* and *USP20* loci, which exhibit representative regulatory signatures (Fig. 1a). In line with previous reports of DNase I hypersensitive sites in exonic regions[7], we found that 3.9% of DNase I hypersensitive sites overlapped exonic regions, underscoring the regulatory potential of exons (Fig. 1b). In parallel, TF ChIP-seq peaks occupy diverse genomic contexts, with the majority of ReMap peaks mapping to intronic (40.9%) and intergenic (25.6%) regions, consistent with traditional enhancers, yet 4.2% of these peaks localise to coding exons (Fig. 1b, Supplementary Table 1). Moreover, exonic regions overlapping classic regulatory features, such as enhancer marks from ENCODE cCREs[14], DNase I hypersensitive sites[15], ATAC-seq peaks (ChIP-Atlas[16]), CAGE TSS (FANTOM[17,18]), and histone modifications like H3K27ac[14] (Fig. 1c, Supplementary Table 1), strengthen existing evidence that some exons could actively participate in transcriptional regulation.

Across the human genome, 74% of coding exons (n = 131,358) overlap at least one TF-binding peak, and 24% were bound by ten or more TFs (n = 42,699), highlighting the frequent occurrence of TF occupancy within coding exons (Fig. 1d). Notably, 8% of these TF-bound exons (n = 13,481) satisfy stringent multi-TF binding thresholds and filtering methods, displaying enhancer-like activity. We therefore designated these 13,481 exons as candidate Exonic Enhancers (cEEs) in the rest of this study. We identified cEEs using stringent criteria to distinguish them from promoters or other regulatory elements while ensuring TF binding was predominantly confined to the exon, which

prevented confounding signals from adjacent intronic regions (Methods). Consequently, some previously tested EEs from the literature[9] were excluded (Supplementary Fig. 1). To evaluate cEEs beyond human, we extended our TF overlap analysis to mouse (*Mus musculus*), fruit fly (*Drosophila melanogaster*), and thale cress (*Arabidopsis thaliana*) (Supplementary Fig. 2-5). Meta-exon analyses show a focused peak of TF summits at the midpoint of these exons, particularly among those with extensive TF co-occupancy (Fig. 1e, Supplementary Fig. 6). In all four species, coding exons show enhancer-like features, including TF occupancy (Supplementary Fig. 7) and chromatin accessibility (Supplementary Fig. 2), indicating that cEEs are evolutionarily widespread (Fig. 1f, M.*mus* cEEs=12,244; *D.mel* cEEs=13,688; *A.tha* cEEs=7,138). Although the exon and gene counts vary across species (Fig. 1g), cEEs are present in nearly all host-gene isoforms (Supplementary Fig. 8), are constitutively spliced across GTEx tissues (i.e., consistently included rather than frequently skipped), and show APPRIS annotations broadly similar to ordinary coding exons (Supplementary Fig. 9, 10). Our cross-species analyses deliver a systematic, multi-species portrait of EEs as a widespread yet still under-appreciated regulatory layer, highlighting the rich complexity embedded within coding exons and reaffirming their pivotal contribution to gene regulation.

### Exonic enhancers display enhancer-associated features across genomes

To assess whether cEEs display canonical enhancer-associated features, we integrated publicly available DNase I hypersensitivity and ATAC-seq datasets from ENCODE[14] and ChIP-Atlas[16]. Across the four species, we observed a high degree of overlap between cEEs and open chromatin marks (Fig. 2a, and Supplementary Fig. 11). Notably, 82% of human cEEs and 67% of mouse cEEs overlapped with DNase or ATAC peaks, and we also detected strong concordance with H3K27ac, H3K4me1 and, in particular, H3K4me2 enrichment (7.4-fold; Supplementary Table 1, Supplementary Fig. 12), a mark characteristic of poised promoters and active enhancers[19,20]. This intermediate methylation state maintains nucleosomes that remain accessible for TF binding yet do not impede RNA-pol II elongation, making H3K4me2 particularly well suited to enhancers embedded within actively transcribed coding exons. Together these chromatin signatures indicate that cEEs adopt an active, TF-permissive configuration, consistent with prior links between DNase/ATAC accessibility and enhancer potential[21].

To functionally validate the enhancer-like activity of cEEs, we compiled a comprehensive multi-species STARR-seq atlas spanning 112 datasets across multiple cell lines and tissues (Fig. 2b, Methods, Supp File). STARR-seq, which quantifies enhancer-driven transcription in a plasmid-based reporter assay[22], revealed millions of potential enhancer peaks, including a discrete subset within cEEs. Across all four species, cEEs overlapped STARR-seq peaks far more frequently than coding exons without TF binding (Ctrl-; Fig. 2c). Their overlap rate matched that of canonical enhancers (Ctrl +), defined as intergenic distal cCREs bound by ≥10 TFs under the same filtering criteria (Methods), implying that cEEs are just as likely to drive reporter expression. Furthermore, metaprofile analyses of STARR-seq data from K-562 (erythromyeloid) and A-549 (respiratory epithelial) cells revealed tissue-specific elevated enhancer signals, notably enriched at exonic enhancers associated with myeloid and respiratory tissues, respectively, compared to control and random genomic regions (Fig. 2d, and Supplementary Fig. 13, Methods).

To validate the regulatory potential of cEEs computationally, we hypothesised that the ChIP-seq signals detected in cEEs might be supported by an enrichment of transcription factor binding sites (TFBS) aligning with the TF peaks, similar to classic enhancers (Ctrl +) (Fig. 2e). Using a genome-wide randomisation strategy, we shuffled TF peaks across each genome assembly and assessed binding motif

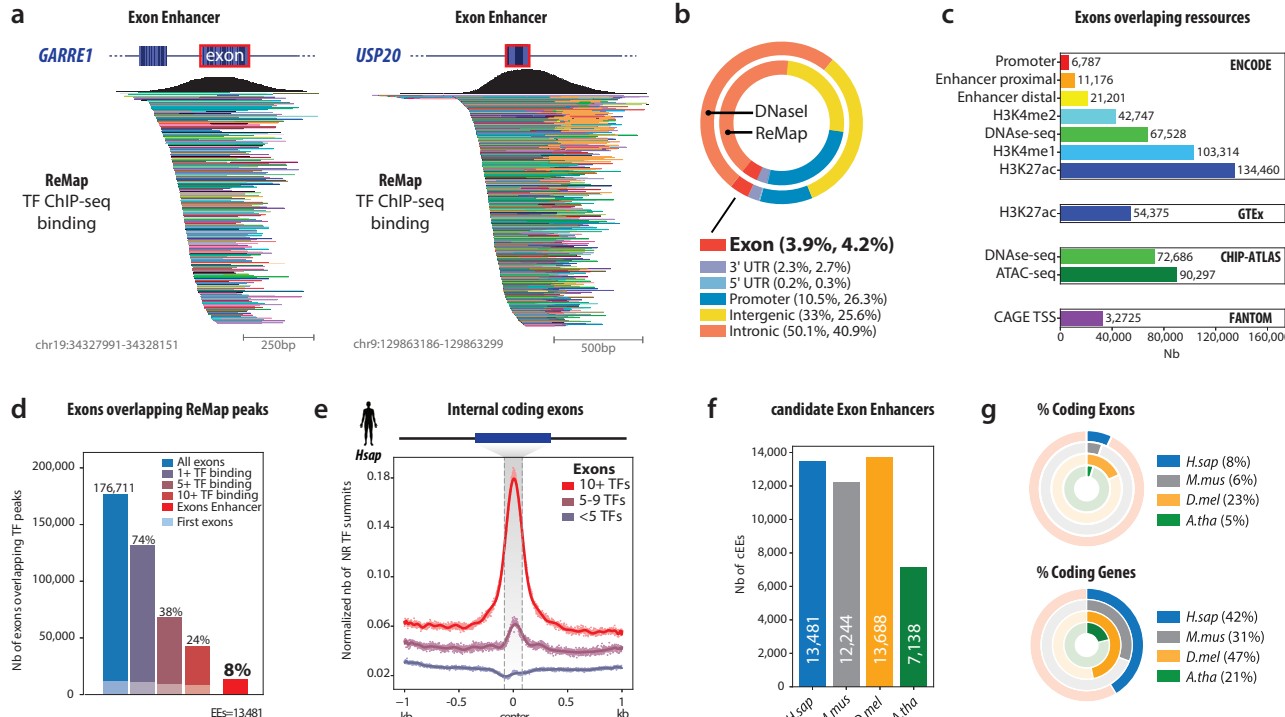

**Fig. 1 | Identification and characterisation of candidate Exonic Enhancers across species. a** Two representative examples of candidate Exonic Enhancers (cEEs) within protein-coding exons (*GARRE1* on human chromosome 19 and *USP20* on chromosome 9). The top black traces depict aggregated TF ChIP-seq signal (ReMap), illustrating a peak of transcription factor (TF) occupancy centred on an internal exon. Each coloured bar below represents a distinct TF dataset from ReMap, collectively underscoring extensive multi-TF binding at these exons. **b** Donut charts showing the proportion of genomic annotations (e.g., intronic, intergenic, promoter, UTR, exon) identified by DNase I hypersensitivity (outer ring) and TF occupancy from ReMap (inner ring). Notably, exons account for ~4% of these regulatory regions, illustrating that a non-negligible fraction of regulatory elements localises to protein-coding exons. **c** Bar plots indicating the overlap of exons with various genomic resources, including ENCODE (e.g., promoter, enhancer, DNase-seq, histone marks), GTEx (H3K27ac histone mark), ChIP-Atlas (DNase-seq, ATAC-seq), and FANTOM (CAGE TSS). A substantial number of exons intersect enhancer-related annotations, implying that coding exons frequently harbour regulatory potential. **d** Histogram depicting the total number of exons overlapping TF-bound peaks according to ReMap, stratified by the minimum number of TFs bound. Of 176,711 exons, 74% have ≥1 TF, 38% have ≥5 TFs, 24% have

≥10 TFs, and 8% are designated as "candidate Exonic Enhancers" (cEEs). This suggests that cEEs, although a minority, are enriched for multiple overlapping TF-binding events. **e** Metagene plot of TF summit density centred on internal coding exons. Exons bound by ≥10 TFs (red trace) exhibit a strong, focused peak at the exon centre compared to exons bound by 5-10 TFs (purple trace) or <5 TFs (grey trace), indicating that highly TF-occupied exons are enriched in regulatory activity precisely at their midpoint. **f** Bar chart showing the number of cEEs detected in four model organisms, human (*H. sapiens*), mouse (*M. musculus*), fruit fly (*D. melanogaster*), and thale cress (*A. thaliana*). The presence of cEEs across diverse species underscores the evolutionary breadth of exon-centred regulatory elements. **g** Donut plots showing the proportion of coding exons (top) and coding genes (bottom) that contain at least one cEE in each species. Percentages are calculated relative to the total number of annotated protein-coding exons (top) or genes (bottom) in the genome. Although *D. melanogaster* has a higher fraction of coding exons (23%) compared to the other species examined, the percentage of cEEs is nonetheless appreciable in all four genomes, suggesting a broadly conserved role for cEEs in gene regulation. Source data are provided as a Source Data file and Zenodo.

occupancy. Remarkably, cEEs maintained a significantly higher occupancy of matching TF with their corresponding binding sites across all four species, compared to randomised conditions (For human, two-sided p = 0; 100,000 randomisations, corroborated by Wilcoxon signed-rank). This pattern, combined with a higher TFBS density in cEEs than negative controls (Supplementary Fig. 14), mirrors the behaviour of classical enhancers and exceeds expectations based on random distributions. These findings establish EEs as a distinct sub-class of enhancers, rather than mere artifacts arising from proximal ligation during chromatin immunoprecipitation, contrasting with earlier interpretations[23]. They also align with the dual role of protein-coding exons[7] as TF-binding platforms thereby advancing the concept of exon-centred regulation.

Previous studies implicated secondary DNA structures in enhancer function[24]. We analysed G-quadruplex (G4) motifs in cEEs versus matched controls (Ctrl- and Ctrl +) across four species. cEEs showed significantly higher predicted G4 sequences and GC content (Fig. 2f, Supplementary Fig. 15), suggesting that G4 formation may enhance TF binding or modulate transcription[25]. In conclusion, our multi-species

analyses confirm that many exons display multiple hallmark characteristics of enhancer, characterised by open chromatin, STARR-seq activity, relevant TF binding sites, and an enrichment of G4 and GC-rich sequences.

## Functional validation of exonic enhancers activity
To experimentally confirm the regulatory potential of cEEs, we selected a panel of candidates identified in our integrative analysis (Figs. 1 and 2) and cloned them into luciferase reporter constructs (Fig. 3a). Each cEE was inserted upstream of a minimal SV40 promoter, allowing quantification of enhancer-driven transcription. In total, we tested 20 cEEs in luciferase reporter assays (Fig. 3a, and Supplementary Table 4). Of these, 8 exhibited moderate to robust transcriptional activation (fold-change range 1.6-11.5) compared to both empty vector and exon-derived control sequences (2 exons without TF, 1 EE previously identified in HepG2[9]). Thus, the enhancer activity detected in vitro is concordant with the in-silico signals of H3K27ac, H3K4me1, concentrated TF occupancy and open chromatin that characterise functional regulatory elements.

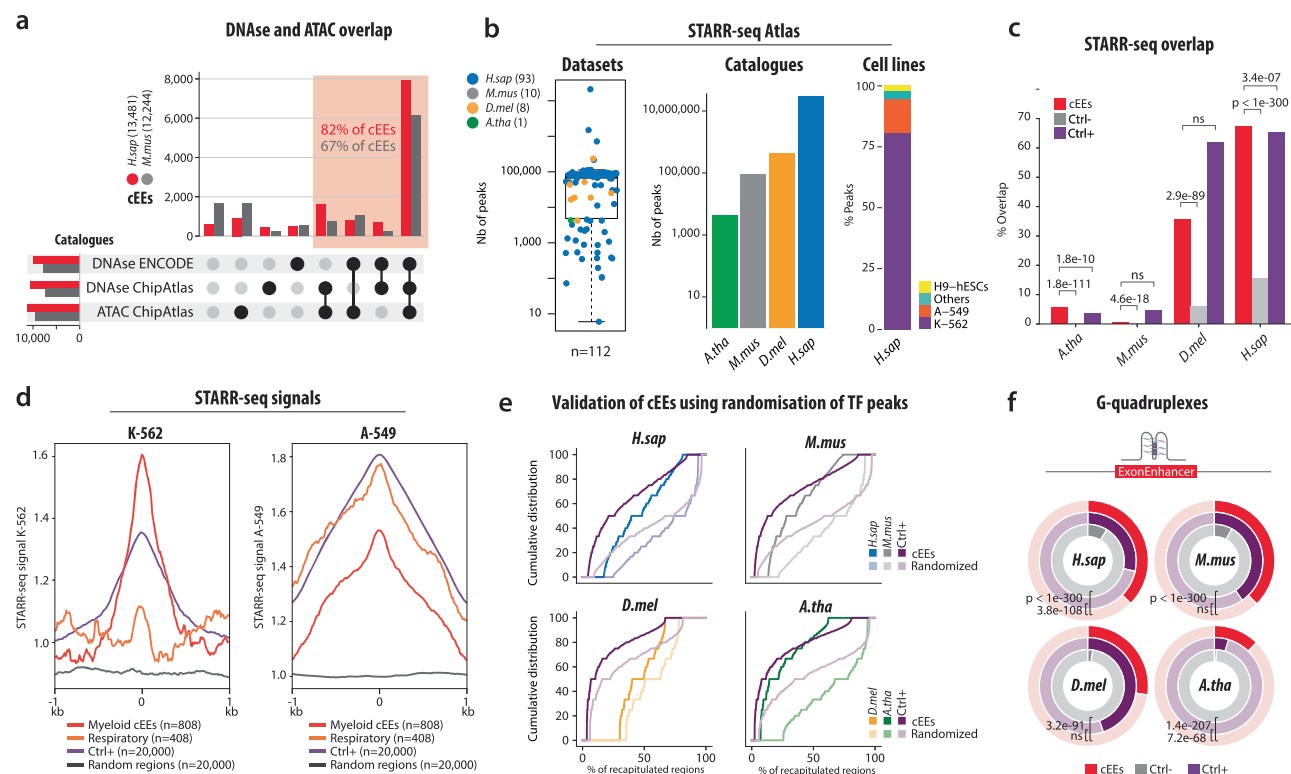

**Fig. 2 | Exonic Enhancers display enhancer-associated features across genomes.**
**a** Combinatorial overlaps of cEEs with DNase I hypersensitive sites and ATAC-seq datasets, derived from ENCODE and ChIP-Atlas resources, for *H. sapiens* (*H.sap*) and *M. musculus* (*M.mus*). The upset plot highlights that 82% of human cEEs and 67% of mouse cEEs exhibit open chromatin signals (red shading). **b** STARR-seq Atlas summary. The box-and-whisker plot (left) illustrates the number of STARR-seq peaks identified per STARR-seq dataset (n = 112 total datasets), colour-coded by species. Box plots show the median (centre line) and interquartile range (box, 25th to 75th percentiles). Whiskers extend to the most extreme values within 1.5×IQR; points beyond the whiskers are plotted as outliers. The middle bar graph depicts the total count of STARR-seq peaks per species, and the stacked bar chart (right) shows cell line or tissue composition in the human dataset (e.g., K-562, A-549, H9-hESCs). **c** Comparison of STARR-seq overlap percentages for cEEs (red), negative controls (Ctrl-, grey), and positive controls (Ctrl + , purple) across the four species. Statistical significance (Fisher's exact one-side test) are indicated, ns = not significant. **d** Metaprofiles depicting STARR-seq signal intensity in human K-562 (left) and A-549 (right) cells, centred on cEEs identified with myeloid (red) and respiratory (orange) signatures. Signals from positive control regions (Ctrl + , purple), negative controls (random genomic regions; dotted grey line) are shown for

comparison. Signal enrichment demonstrates significant enhancer activity specifically around tissue-associated cEEs. **e** Validation of cEEs using randomisation of TF peaks. Cumulative distribution curves compare the fraction of candidate exonic enhancers (cEEs, red) and positive controls (Ctrl + , purple) overlapping transcription factor (TF) peaks relative to randomised TF peak positions (light lines) for each species. In all species, cEEs show significantly higher TF binding than expected by chance. Statistical significance was assessed using two-sided Wilcoxon signed-rank tests on paired differences between observed and randomised values (H. sapiens cEEs, $p < 1 \times 10^{-300}$; M. musculus cEEs, $p < 1 \times 10^{-300}$; D. melanogaster cEEs, $p = 1.98 \times 10^{-77}$; A. thaliana cEEs, $p < 1 \times 10^{-300}$; corresponding positive controls all $p < 2.2 \times 10^{-16}$). As a complementary nonparametric validation, two-sided paired sign-flip permutation tests on the mean difference were performed using 100,000 permutations. **f** G-quadruplex (G4) content within cEEs (red arcs), negative controls (grey arcs), and positive controls (purple arcs) across *H. sapiens*, *M. musculus*, *D. melanogaster*, and *A. thaliana*. The doughnut plots highlight significantly higher G4 occurrences in cEEs relative to controls (p-values shown on the plot, ns = not significant, Fisher exact one-sided test). Source data are provided as a Source Data file and Zenodo.

To investigate the effect of selected cEEs (e.g., EE3 and EE19) on promoters of host and distal genes, we integrated promoter capture Hi-C (PCHi-C)[26] and ENCODE-rE2G[27] datasets (Fig. 3b, c, and Supplementary Table 5). PCHi-C mapping shows that each cEE lies in chromatin domains contacting its host-gene promoter and distal promoters, indicating long-range spatial proximity while, at the current 25kb-2Mb resolution, not ruling out contributions from nearby regulatory elements.

To determine if these cEEs can activate their contacted promoters, we replaced the standard SV40 promoter in our luciferase constructs with the predicted target promoters (P1, P2, P3) of two of our strongest cEEs (EE3 and EE19). Reporter assays demonstrated a significant increase in activity for all three EE3 and EE19 associated promoters, supporting the notion that these cEEs can engage multiple promoters across substantial genomic distances. Consistent with promoter independent behaviour expected of canonical enhancers, EE19 was at least as active, and marginally more active, when paired with the weak SV40 minimal promoter, which is well known to respond

strongly to heterologous enhancer input. These findings indicate that EEs can directly engage their endogenous promoters over substantial genomic distances, underscoring their role in long-range gene regulation.

## Coding variants influencing exonic enhancer function

We next investigated how naturally occurring genetic variation affects cEEs activity. Using gnomAD[28], we examined synonymous, missense, and loss-of-function substitutions that overlapped predicted TFBS. At two representative loci, *GARRE1* (EE3) and *USP20* (EE19), we detected synonymous and missense variants within TF consensus sites, as predicted by JASPAR[29]. Reporter luciferase assays revealed that these variants disrupted enhancer activity, leading to a significant reduction or complete loss of function (Fig. 3d,e, Supplementary Table 6).

Conversely, in silico introduction of synonymous variants designed to create or strengthen TF-binding motifs within the EE20 sequence produced an increase in luciferase signal (Fig. 3f). Hence, even ostensibly 'silent' mutations in coding regions can have a

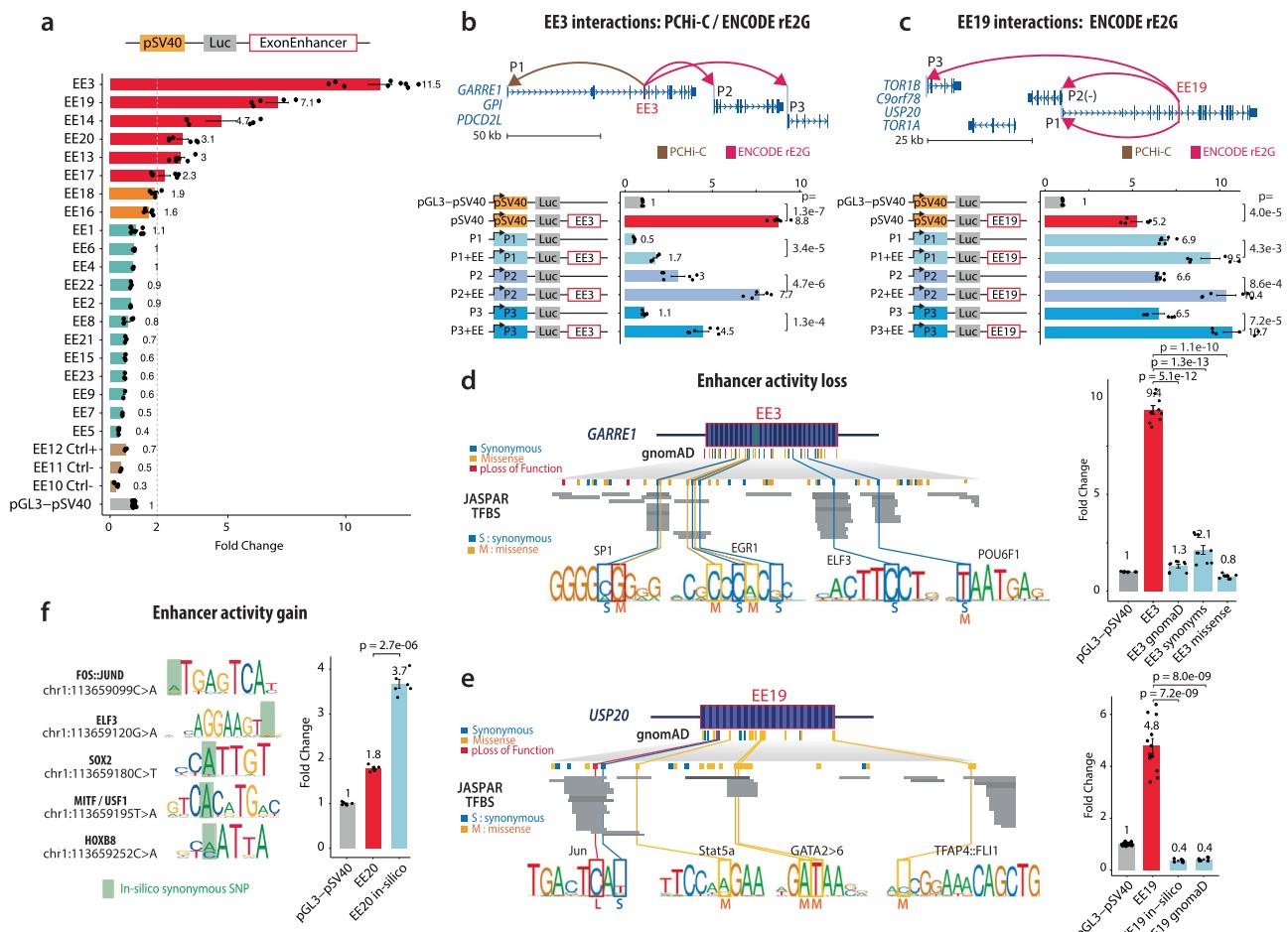

**Fig. 3 | Functional characterisation of cEEs and the impact of genetic variants.**
(**a**) Luciferase reporter assays for a panel of cEEs cloned upstream of the SV40 promoter. Bars show relative fold-change in luciferase activity compared to the empty pGL3-pSV40 vector; control constructs are indicated (Ctrl). The number of measurements per construct corresponds to the number of biological replicates performed (EE3 and EE1: n = 9; EE8, EE5, EE20, EE19, EE18, EE17, EE16, EE14, EE13, EE10: n = 6; EE9, EE7, EE6, EE4, EE23, EE22, EE21, EE2, EE15, EE12, EE11: n = 3). Each data point represents one independent biological measurement. Error bars indicate mean ± SEM. across biological replicates. Strongly active constructs were tested with additional biological replicates, whereas low-activity constructs were analysed using three biological replicates. (**b**) PCHi-C and ENCODE-rE2G analyses reveal physical interactions between EE3 and nearby gene promoters (*GARRE1, GPI, PDCD2L*). The schematic (top) depicts chromatin looping (arcs) linking EE3 to three putative promoters (P1-P3). The bar chart (bottom) shows luciferase activities of combinations of EEs and promoters, indicating synergistic enhancer effects. Two-sided Student's t-test; exact p-values are shown (b-f). Data are shown from biological experiments, with each data point representing one biological replicate (n = 6 total). Error bars indicate the mean ± SEM. across biological replicates. (**c**) Similarly, EE19 engages promoters of *TOR1B, C9orf78, USP20*, and *TOR1A* through long-range chromatin interactions (top). Luciferase assays (bottom) recapitulate these contact-based enhancer-promoter relationships, further supporting the regulatory potency of EE19. Error bars in all panels represent mean ± SEM. for at least three biological replicates. EE19 is slightly stronger with SV40 than with its native

promoter (paired t = 2.46, p = 0.047), as expected for SV40's deliberately weak, enhancer-responsive design. Data are shown from biological experiments, with each data point representing one biological replicate (n = 6 total). Error bars indicate the mean ± SEM. across biological replicates. **d, e** Representative examples of genetic variants affecting EE function in the *GARRE1* (EE3) and *USP20* (EE19) loci, respectively. The schematics depict gnomAD variant classes; synonymous (S), missense (M), and loss-of-function (L); along with predicted transcription factor (TF) binding sites based on JASPAR motifs. The bar plots at right show the resulting luciferase activity of reference and variant EE constructs relative to pGL3-pSV40. Constructs either have only missenses, synonymous or a mix of the several variant classes. EE19 in-silico construct consists of synonymous mutations. Data are shown from biological experiments. For EE3, n = 8 biological replicates and for EE19, n = 5 biological replicates; other constructs were tested using n = 6 biological replicates. Each data point represents one independent biological experiment. Error bars indicate the mean ± SEM. calculated across biological replicates. **f** Variants that confer enhancer activity gain in selected TF binding sites. Sequence logos illustrate key TF motifs (e.g., FOS::JUND, EBF3, SOX9, MTF1/USF1, HOXB), with in silico synonymous SNPs highlighted in green. Fold-change data compare EE-driven reporter activity against pGL3-pSV40. Data are shown from independent biological experiments, with constructs tested using n = 6 biological replicates. Each data point represents one biological measurement (n = 6 total). Error bars indicate the mean ± SEM. calculated across biological replicates. Source data are provided as a Source Data file and Zenodo.

regulatory impact by altering enhancer function. By coupling mutational data with functional assays, we demonstrate that EEs can acquire gain- or loss-of-function phenotypes in the presence of coding mutations. Collectively, these observations position EEs as not only important cis-regulatory elements in normal gene expression but also as potential contributors to pathological states when harbouring disruptive variants.

## Population genetics and selection on exonic enhancers

To explore how coding variation shapes exonic-enhancer function, we first interrogated population and evolutionary data for cEEs. Using gnomAD[28], we observed that cEEs carry rarer variants and harbour a higher synonymous fraction, yielding a lower missense-to-synonymous ratio (pN/pS = 0.59 vs 0.62; $\chi^2$ $p = 1.3 \times 10^{-137}$) and more synonymous changes (0.33 vs 0.27; Mann-Whitney $p < 1 \times 10^{-16}$;

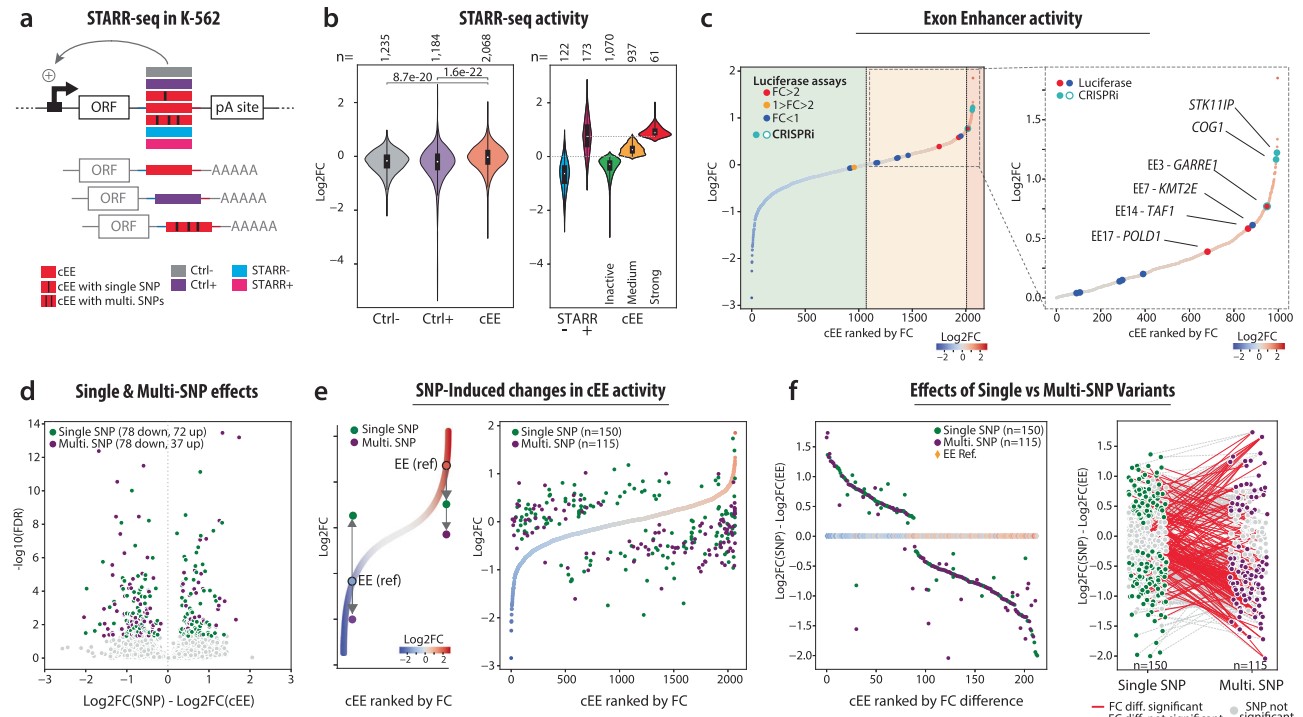

**Fig. 4 | STARR-seq analysis of cEEs and the impact of single and multiple SNPs on enhancer activity. a** Schematic overview of the STARR-seq experimental design in K-562 cells. cEEs (red) and control sequences (Ctrl-, purple; Ctrl +, grey, STARR+ dark red, STARR- dark green) were cloned downstream of a constitutively active promoter driving a reporter ORF and upstream of a poly(A) signal. Both synonymous single-SNP and multi-SNP variants were assessed. **b** Violin plots showing the distribution of STARR-seq activity (log2 fold change) for negative controls (Ctrl-), positive controls (Ctrl +), and cEEs. STARR± indicates previously identified regions with weak and strong STARR-seq activity. STARR-seq activity classes ("Strong", "Moderate", "Weak") reflect log₂FC thresholds. The "Strong" class represents a deliberately stringent cutoff to capture only the highest enhancer outputs and constitutes a minority in all categories, consistent with prior STARR-seq studies. Pairwise two-sided t tests with Bonferroni correction were performed. P-values are shown on the figure. **c** Rank-ordered plot of cEEs by STARR-seq fold change, with colour-coded luciferase outcomes (red, orange, blue). The right-hand zoom highlights top-ranked cEEs and cEEs tested by luciferase reporter assays labelled by the associated gene locus (e.g., *MAP3K13, SERTAD2*, EE3-*GARRE1*, EE7-*KMT2E*, EE14-*TAF1*, EE17-*POLD1*). **d** Scatterplot of variant effects. Single-SNP (green) and multi-

SNP (purple) cEEs are plotted based on the difference between log2 fold change of the reference cEE and variant cEE on the x-axis, and -log10(FDR) (padj <0.05, Benjamini and Hochberg), on the y-axis. Points above the grey background indicate significant divergence in enhancer activity linked to SNP presence. For panels d-e-f, two-sided Wald tests with Benjamini−Hochberg FDR correction (padj<0.05) were used. P-values provided in the supplementary source file and Zenodo. (**e**) Right panel: rank-ordered distribution of cEEs by STARR-seq fold change. Reference cEEs are contrasted with variant cEEs carrying single (green) or multiple (purple) SNPs. (**f**) Left panel: difference in log2 fold change (EE[SNP] - EE[ref]) is plotted for single- and multi-SNP subsets, indicating the extent to which variant alleles modify enhancer activity. Right panel: Comparison of STARR-seq activity changes for single- vs. multi-SNP cEEs, shown as log2 fold-change (EE harbouring SNP - EE reference) on the y-axis. Green circles denote cEEs with single-SNP variants (n = 150), purple circles denote EE with multi-SNP variants (n = 115). Connecting lines link cEEs harbouring both single- and multi-SNP variants; red lines indicate significant differences (padj <0.05, two-sided Wald tests with Benjamini-Hochberg FDR correction), and grey lines indicate nonsignificant differences. Source data are provided as a Source Data file and Zenodo.

Supplementary Fig. 16; Supplementary Table 2). Stratifying by allele-frequency bins shows a clear skew toward ultra-rare variants in cEE (AF ≤ 0.01%: 90.93% vs 90.44%; two-proportion Z-test $p = 1.82 \times 10^{-13}$; $\chi^2$ across bins $p = 2.5 \times 10^{-31}$; Supplementary Table 3). While SNP density increases with TF-binding load, the allele-frequency spectrum is essentially unchanged across TF deciles (~91% ultra-rare, 6% rare, 2% low-frequency, 1% common; Supplementary Table 3), indicating that purifying selection acts broadly across cEEs rather than being confined to the most TF-dense exons (Supplementary Fig. 21–22). cEEs also display a slightly higher human-mouse dN/dS together with positive values for the Direction-of-Selection statistic (DoS) and the Fixation Index (FI), metrics that compare divergence to polymorphism, suggesting that compensatory coding changes have accumulated over deep evolutionary time (Supplementary Table 2 and Supplementary Fig. 21-22). Overall, these results indicate that cEEs appear constrained at the population level yet evolve via compensatory substitutions over deep time. Given these constraints, we next quantified cEE activity and variant effects at scale using STARR-seq.

## STARR-seq validates hundreds of Exonic Enhancers

We previously observed that many cEEs overlapped with publicly available STARR-seq predictions, yet those predicted enhancers often included adjacent intronic regions, obscuring whether the exonic sequence alone was responsible for enhancer activity. To assay genome-wide enhancer activity of cEEs, we screened a STARR-seq library of 2,068 exon-bounded fragments (no intronic sequences) in K-562 cells alongside positive and negative controls (Fig. 4a, Methods), ensuring that any signal originates from coding DNA rather than neighbouring intronic elements. This assay positions cEE candidate elements downstream of a minimal promoter, enabling direct and quantitative measurement of enhancer-driven transcription[22]. Across replicates, K-562 cEEs exhibited significantly higher STARR-seq activity than negative controls (Ctrl-, n = 1235, $p = 8.74 \times 10^{-20}$ two-sided t-test) and, in many cases, approached or exceeded the activity levels of established classic enhancers (Ctrl +, n = 1184, $p = 165 \times 10^{-22}$ two-sided t-test) (Fig. 4b). cEEs associated with A-549 or GM12878 cell line signatures show significantly lower activity in K-562 cells, suggesting context-dependent regulatory function (Supplementary Fig. 17). Used

only as a host-gene proxy (not direct enhancer output), complementary GTEx/ENCODE transcriptomes analyses indicate a mixed landscape: cEEs associate with both broadly and narrowly expressed genes, paralleling canonical enhancer behaviour (Supplementary Figs. 18–20). Together, these results reveal that a substantial fraction of coding DNA harbours strong enhancer activity, further highlighting its regulatory potential.

To define thresholds for cEE activity, we used controls such as promoter elements previously characterised with enhancer-like activity (STARR + , $n = 173$), and their inactive counterparts (STARR-, $n = 122$), validated through prior STARR-seq studies[30]. Based on these benchmarks, we categorised cEEs into three groups: inactive cEEs (green, fold change [FC] ≤ 0), moderately active cEEs (orange, 0 < FC <median STARR + ), and highly active cEEs (red, FC ≥ median STARR + ) (Fig. 4b). Among 2068 exon-bounded inserts, 61 cEEs (2.9%) reached the 'Strong' activity bin, compared with 21 of 1184 Ctrl+ enhancers (1.8%), indicating that high enhancer potency is not diminished within coding exons. We observed 9 cEEs previously validated via luciferase assays (Fig. 3a) exhibited STARR-seq signals which are globally consistent with these activity thresholds (Fig. 4c). Top cEEs (FC>2x) fall in the upper tail of the STARR-seq rank distribution, indicating qualitative concordance between assays. Among the top-tier cEEs, we identified loci linked to essential cellular processes, including signal transduction (*MAP3K13*), transcriptional regulation (*TAF1*), and DNA replication (*POLD1*) (Fig. 4c). These results establish EEs as key cis-regulatory elements capable of driving robust transcriptional activity within coding regions.

## Coding variants modulate exonic enhancers activity

To assess whether natural genetic variation can modulate EE activity, we introduced single-nucleotide polymorphisms (SNPs) from gnomAD predicted to alter TFBS, into our cEE STARR-seq library (Fig. 4a). We tested single-SNP cEEs with synonymous variants as well as multi-SNPs cEEs with diverse functional impacts. STARR-seq measurements revealed that some variant cEEs deviated from their reference sequence activity, indicating that coding-region mutations can alter enhancer output (Fig. 4d, e). The volcano plot (Fig. 4d) shows that cEEs carrying multiple SNPs often exhibit a reduced enhancer activity ($n = 78$ down, $n = 37$ up), presumably due to the cumulative disruption of TF-binding motifs or other regulatory features. Meanwhile, rank-ordered analyses (Fig. 4e) confirm that both single- and multi-SNP cEE variants diverge from the cEE reference activity to varying extents, suggesting that coding-region variations can influence enhancer activity, as demonstrated at a lower scale (Fig. 3b–d). Strikingly, variants found in cEEs with high activity tend to reduce reporter output, whereas those in cEEs with low or no basal activity often increase it (Fig. 4f). Simulations confirmed that this negative correlation far exceeds what would be expected from a regression to the mean effect (Supplementary Fig. 23).

Likewise, comparisons of $\log_2$ fold-change between reference cEEs and their mutated counterparts (single- or multi-SNP) illustrate a broad spectrum of activity alterations (Fig. 4f, left). In particular, multi-SNP variants show more pronounced deviations from the reference enhancer activity than their single-SNP variants counterparts (Fig. 4f, right), emphasising how multiple concurrent mutations can amplify changes in enhancer function. Combining high-throughput STARR-seq with targeted SNP mutagenesis, we demonstrate that while EEs can drive robust transcriptional activity, they are remarkably sensitive to even minor coding-region variants, underscoring both their role as essential cis-regulatory platforms and their vulnerability to mutations that may lead to functional or pathological consequences[12].

## Exonic Enhancers form robust enhancers-target gene interactions

Having established that cEEs can function as bona fide enhancers using STARR-seq (Fig. 4), we next sought to identify their potential target genes. To this end, we generated an interaction atlas by integrating three complementary datasets: (i) a compiled promoter capture Hi-C collection[26], (ii) enhancer-promoter regulatory interactions from the ENCODE-rE2G resource[27] and (iii) eQTL associations from GTEx[31]. We focused on overlap between cEEs and these interaction signals, comparing them with both classic enhancers (Ctrl + ) and normal coding exons (Ctrl-). Overall, cEEs exhibited broader and more frequent overlaps with at least one of the three datasets, comparable to classic enhancers and substantially exceeding the negative controls even at cellular or tissue contexts (Fig. 5a, and Supplementary Fig. 24). We categorised cEE-target gene interactions according to whether the target gene was the cEE host gene (internal), located elsewhere in the genome (external), or both (mixed). To increase the reliability of these associations for subsequent analyses, we restricted analyses to robust cEE-target gene pairs (cEE-TG) supported by at least two of the three resources (promoter capture Hi-C, ENCODE-rE2G, or GTEx eQTL). Notably, 26% of cEEs showed such robust interactions, compared with 35% for classic enhancers and only 2% for negative controls (Fig. 5b). These findings underscore that EEs not only display enhancer-like regulatory potential (Figs. 1–4) but also form stable interactions with downstream targets.

## Genetic variation in exonic enhancers impact cancer-related expression

We next addressed whether variants in cEEs could have pathological or functional outcomes in human cancers. To capture this variation at scale, we utilised the TCGA Pan Cancer Atlas[32] comprising over 10,000 tumour samples across 33 cancer types (Fig. 5c). To evaluate whether cEE variants influence the expression of their putative target genes in a pathological context, we intersected the PanCanAtlas variants located in cEEs with the robust cEE-TG pairs previously defined (Fig. 5d). Across the 33 tumour types, the number of cEE variants per patient ranged from a single event to thousands, highlighting the broad heterogeneity of mutational burden in cancer (Fig. 5d left). For each variant category, all variants (yellow) and synonymous variants only (green), we performed differential expression analyses of the corresponding target genes, visualized as flattened volcano plots (Fig. 5d right). Notably, a subset of cEE-target gene pairs exhibited significant transcriptional changes, suggesting that cEE mutations may disrupt cis-regulatory architecture to drive oncogenic or impair tumour-suppressive pathways. Importantly, even synonymous substitutions (green), historically deemed 'silent', were associated with altered expression (Fig. 5d right), demonstrating that such variants can subtly modulate enhancer activity. Across cancer types, TCGA variants intersecting cEE frequently altered TF binding (Fig. 5e), with the effects ranging from weak (light red) to strong (dark red). Notably, synonymous mutations alone also impacted TFBSs (Fig. 5e, right), reinforcing their regulatory significance independent of coding changes. To illustrate our prediction of TF binding alterations for cEEs (Supplementary Fig. 25), we showcase a cEE of the *MMRN2* gene (Supplementary Fig. 26), with predicted changes in TF binding affinity visualized via a public UCSC trackhub (Methods). Collectively, these findings indicate that a subset of candidate exonic enhancers are somatically mutated in tumours and that such mutations are statistically associated with altered expression of nearby targeted genes. Although causal links to tumour progression remain to be demonstrated experimentally, these observations highlight cEEs as genomically and transcriptionally responsive loci in cancer.

## Prognostic relevance of exonic enhancers-target gene disruption

To evaluate whether cEE-mediated regulatory changes may have clinical relevance, we selected examples of differentially expressed target genes identified in four cancer types: lower-grade glioma (LGG), lung adenocarcinoma (LUAD), stomach adenocarcinoma

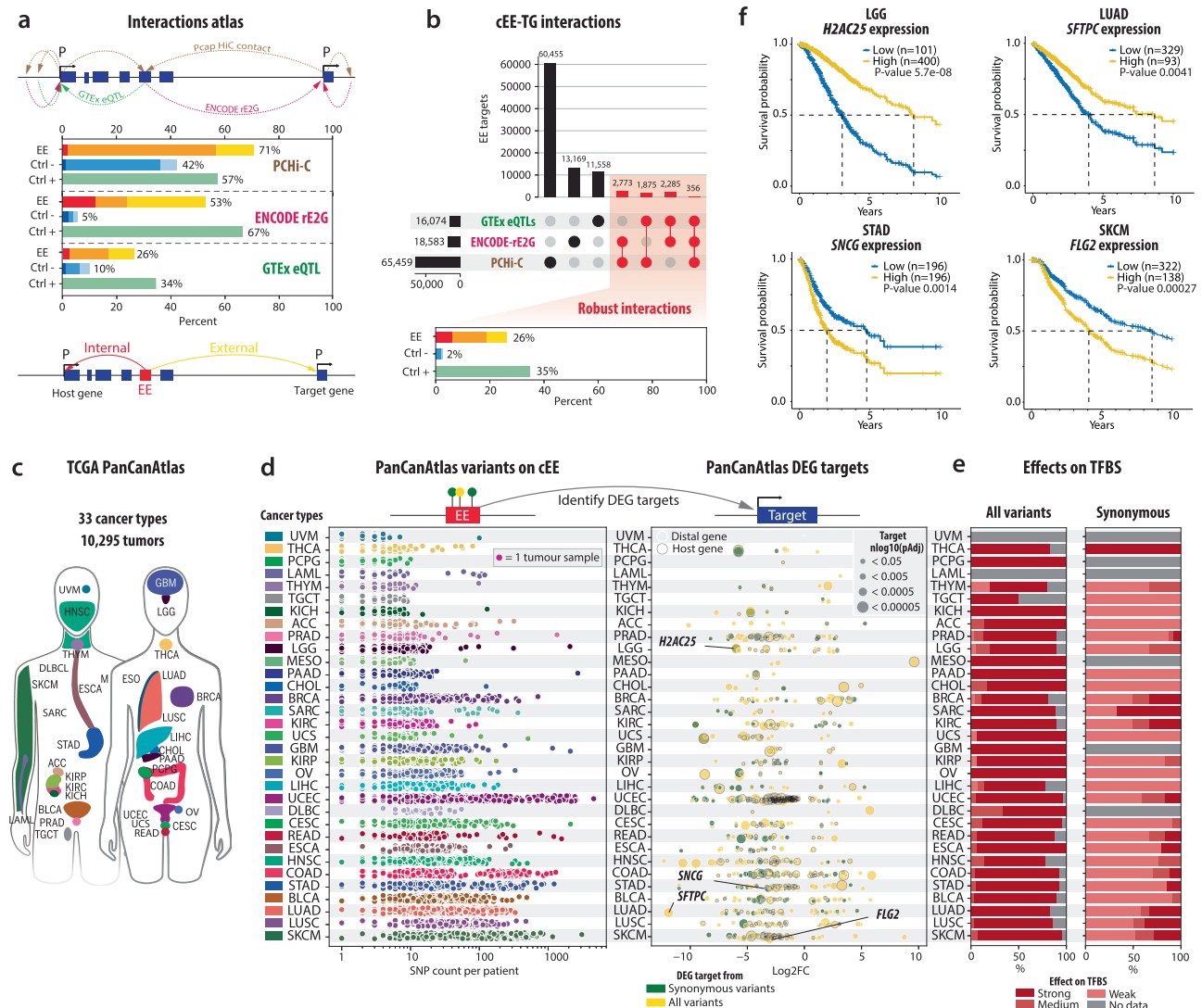

**Fig. 5 | Pan-cancer analysis of candidate Exonic Enhancers and their target gene interactions. a** Construction of an "interaction atlas" to map potential cEE target genes, drawing on promoter capture Hi-C data from Laverré et al. [26], ENCODE-rE2G from Gschwind et al. [27], and GTEx eQTLs [50]. The schematic shows internal (red), external (yellow), and mixed (orange) cEE-target gene configurations, with bar plots indicating that cEEs exhibit robust overlaps in these data sets, similar to positive controls (Ctrl +, green) and exceeding negative controls (Ctrl-, blue). **b** For downstream analyses, only cEE-target gene (cEE-TG) interactions confirmed by at least two of the three resources were retained, yielding 26% robust interactions for cEEs, compared to 35% for Ctrl+ enhancers and 2% for Ctrl- sequences. **c** Overview of the TCGA PanCanAtlas dataset; encompassing 33 cancer types (~10,000 tumours); used to explore possible pathogenic effects of coding variants within cEEs. Cancer sites are colour-labelled based on prior nomenclature. Panel c is adapted from Corces et al. [68] and reused with permission from AAAS. The figure was modified to include additional cancer types. **d** Left panel: distribution of PanCancer variants per patient located within identified cEEs across various tumour types. Right panel: corresponding differentially expressed target genes (DEGs) identified

through cEE-TG interactions. Each row represents a flattened volcano plot, illustrating expression changes in target genes, with DEGs linked to synonymous variants shown in green and those linked to all variants shown in yellow. DESeq2 two-sided Wald tests with FDR correction were used. **e** Predicted impact of PanCancerAtlas variants (left) and synonymous variants (right) on TFBSs within cEEs. Disruptive effects were classified as strong (dark red), medium (red), or weak (light red) based on FABIAN-predicted binding disruption scores using JASPAR TFBS motifs. Grey bars indicate cancer types with no available data, or non-computed disruption score due to the variant type. **f** Survival curves showing differential overall survival (OS) based on the expression levels of four example cEE-regulated target genes (*H2AC2S*, *SFTPC*, *SNCG*, and *FLG2*) in LGG, LUAD, STAD, and SKCM, respectively. High-expression (yellow) and low-expression (blue) cohorts demonstrate significant survival differences, highlighting the potential prognostic value of these cEE target genes. Two-sided log-rank tests were used to compare Kaplan–Meier survival curves between median-split expression groups; hazard ratios were estimated using Cox proportional-hazards models. Source data are provided as a Source Data file and Zenodo.

(STAD), and skin cutaneous melanoma (SKCM). For each of them, we assessed the prognostic value of high or low target-gene expression (Fig. 5f). In all four cancers, altered expression correlated with differential overall survival, signifying that variants located within cEEs can not only perturb enhancer activity but may also drive meaningful shifts in tumour cell biology. Although these correlations suggest a role for EE disruptions in tumorigenesis, direct in vivo validation lies beyond our scope, so the link remains primarily inferential.

Additionally, cEEs harbour a higher fraction of disease- and measurement-related GWAS variants than controls and are enriched for pleiotropic SNPs that exhibit robust regulatory interactions (Supplementary Fig. 27). These results emphasise that exonic enhancer elements appear to be highly sensitive to natural and pathological DNA variation, with potential consequences for cancer development and patient outcomes.

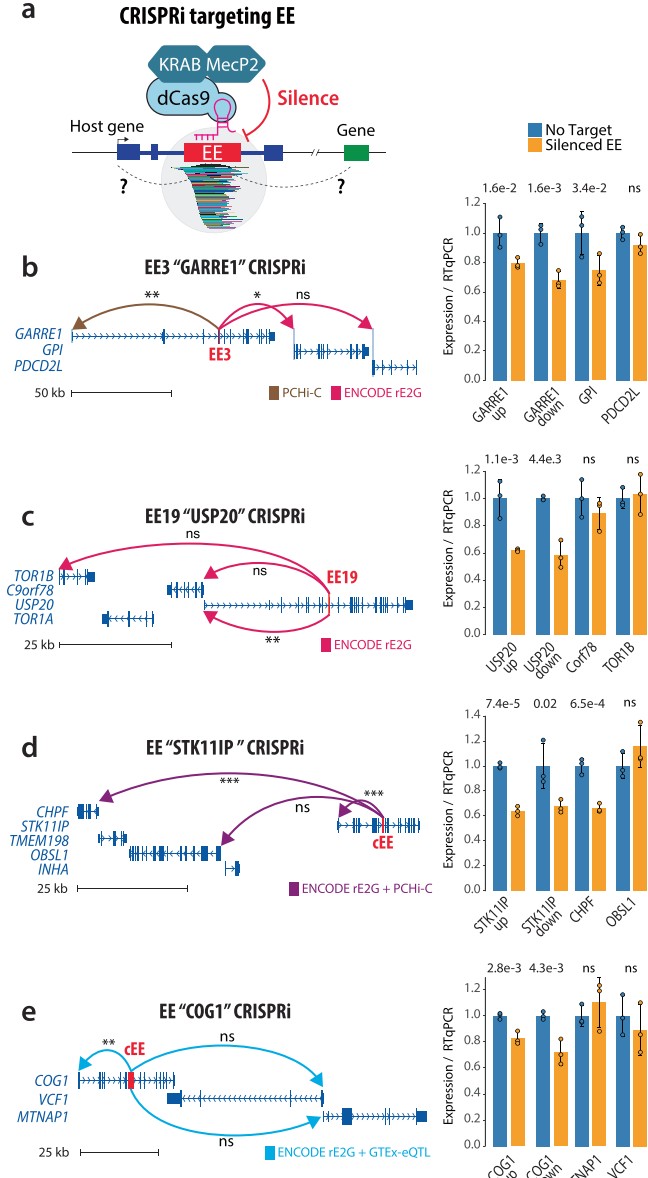

**Fig. 6 | CRISPRi silencing validates EE-mediated gene regulation. a** Schematic of the CRISPR interference approach using dCas9-KRAB-MeCP2 targeted to cEEs. **b**–**e** Four representative cEEs (EE3 *GARRE1*, EE19 *USP20*, cEE *STK11IP*, and cEE *COG1*) are shown in their genomic context, each predicted to contact its host or distal target genes via one or more interaction datasets (promoter capture Hi-C, ENCODE-rE2G, and GTEx). Arcs denote these predicted regulatory interactions. Bars show relative mature mRNA levels quantified by exonic RT-qPCR and compare measurements of gene expression in no-target controls (blue bars) versus CRISPRi-mediated EE silencing (yellow bars). cEEs validated by both luciferase assays and STARR-seq (EE3 *GARRE1*, EE19 *USP20*) exhibit reduced transcription of both their host genes and predicted targets (b,c). Similarly, silencing of top-ranked cEEs from STARR-seq (cEE *STK11IP*, cEE *COG1*) diminishes expression of associated loci (**d**, **e**). Error bars represent mean SEM. (*n* = 3 biological replicates). Statistical significance was assessed using a one-sided unpaired two-sample Student's t-test. Exact p-values are shown above the bars; ns not significant. Source data are provided as a Source Data file and Zenodo.

## CRISPRi silencing validates exonic enhancers-mediated gene regulation

To assess whether cEEs actively modulate gene expression in vivo, we employed a potent CRISPR interference system (dCas9-KRAB-MeCP2) targeting four selected cEEs in K-562 cells (Fig. 6a). Each cEE was previously implicated as a regulatory hub in our interaction catalogue,

contacting one or more host or distal target genes via promoter capture Hi-C, ENCODE-rE2G and GTEx eQTL mapping. We monitored the effects of cEE silencing on the mature mRNA level of the host (placing primers upstream and downstream of each cEE to capture potentially truncated transcripts) and neighbouring genes by RT-qPCR (Methods; Supplementary Table 7-8). In all four cases we observed a regulatory effect of the cEE inactivation, whereas matched gRNAs targeting non-cEE coding exons from the same genes (exons lacking obvious enhancer signatures) did not alter expression (Supplementary Fig. 28), supporting a sequence-specific enhancer effect. Notably, repression of EE3 *GARRE1* and EE *STK11IP* resulted in significant reduction of the host and one of the distal interacting genes (*GPI* and *CHPF*, respectively) (Fig. 6b, d), while repression of EE19 *USP20* and EE *COG1* resulted in significant reduction of the host gene only. (Fig. 6c, e). These results demonstrate that EEs can contribute to long-range control of gene expression in the mammalian genome.

## Evolutionary dynamics and selective pressure on Exonic Enhancers

To determine whether cEEs represent ancient, well-established regulatory elements or arise sporadically in the genome, we assessed their conservation patterns across multiple mammalian lineages and beyond. Pairwise genome alignments between human and mouse revealed that 28% of human-defined cEEs map to mouse-defined cEEs (Fig. 7a), indicating that a notable subset is evolutionarily retained. The remaining 70% appear not-shared with cEEs identified in mouse, potentially reflecting lineage-specific regulation or incomplete annotation arising from differences in TF coverage across species. Remarkably, even the mouse exons aligned to these non-shared human cEEs still display a low TF occupancy exceeding negative controls, suggesting emergent or sub-threshold enhancer potential. These putative orthologous cEEs could represent ancestral exonic enhancers that have partially decayed or remain undetected in some lineages, reflecting incomplete TF coverage or lineage-specific expansions. For comparison, we examined classic enhancers (Ctrl + ), of which only 11% overlapped conserved enhancer loci in mouse, consistent with partial evolutionary retention of enhancer elements[33] (Supplementary Fig. 29). Collectively, these findings suggest that cEEs can follow both lineage-specific and conserved regulatory trajectories within coding sequences.

To further assess cross-species conservation, we incorporated multi-species ChIP-seq data[34] of four key TFs (CEBPA, FOXA1, HNF4A, and ONECUT1) in liver across human, macaque, mouse, rat, and dog. By intersecting cEEs with these multi-genome TF ChIP-seq datasets and projecting them onto multiple alignments, we identified coding exons that exhibit TF binding in multiple species (Fig. 7b, c), including some cEEs with conserved occupancy across three or more lineages. In Fig. 7d, we compare the proportion of cEEs shared between human and other species using both our dataset (red asterisk for human-mouse) and the four-TF liver data (coloured dots). While the absolute percentages differ due to TF set size and experimental scope, the multi-species results recapitulate a phylogenetic trend: cEEs are more frequently shared with macaque than with rat or dog. This pattern is also seen for intergenic enhancers and previously defined cis regulatory modules (CRMs). Together, these comparisons highlight that a subset of cEEs is evolutionarily conserved across mammals.

We next examined the nucleotide-level evolutionary conservation of cEEs, both globally and stratified by TF occupancy (Fig. 7e). PhyloP scores from 100-species alignments[35] revealed that cEEs display significantly higher sequence constraint than control coding exons (Ctrl-) or typical enhancers (Ctrl + ) ($p < 2.2 \times 10^{-16}$, Wilcoxon rank-sum test). Moreover, cEEs bound by fewer TFs tend to be more strongly conserved and associate with evolutionarily ancient genes with less variants, whereas those with higher TF occupancy exhibit signs of accelerated evolution, have more variants and associate with more

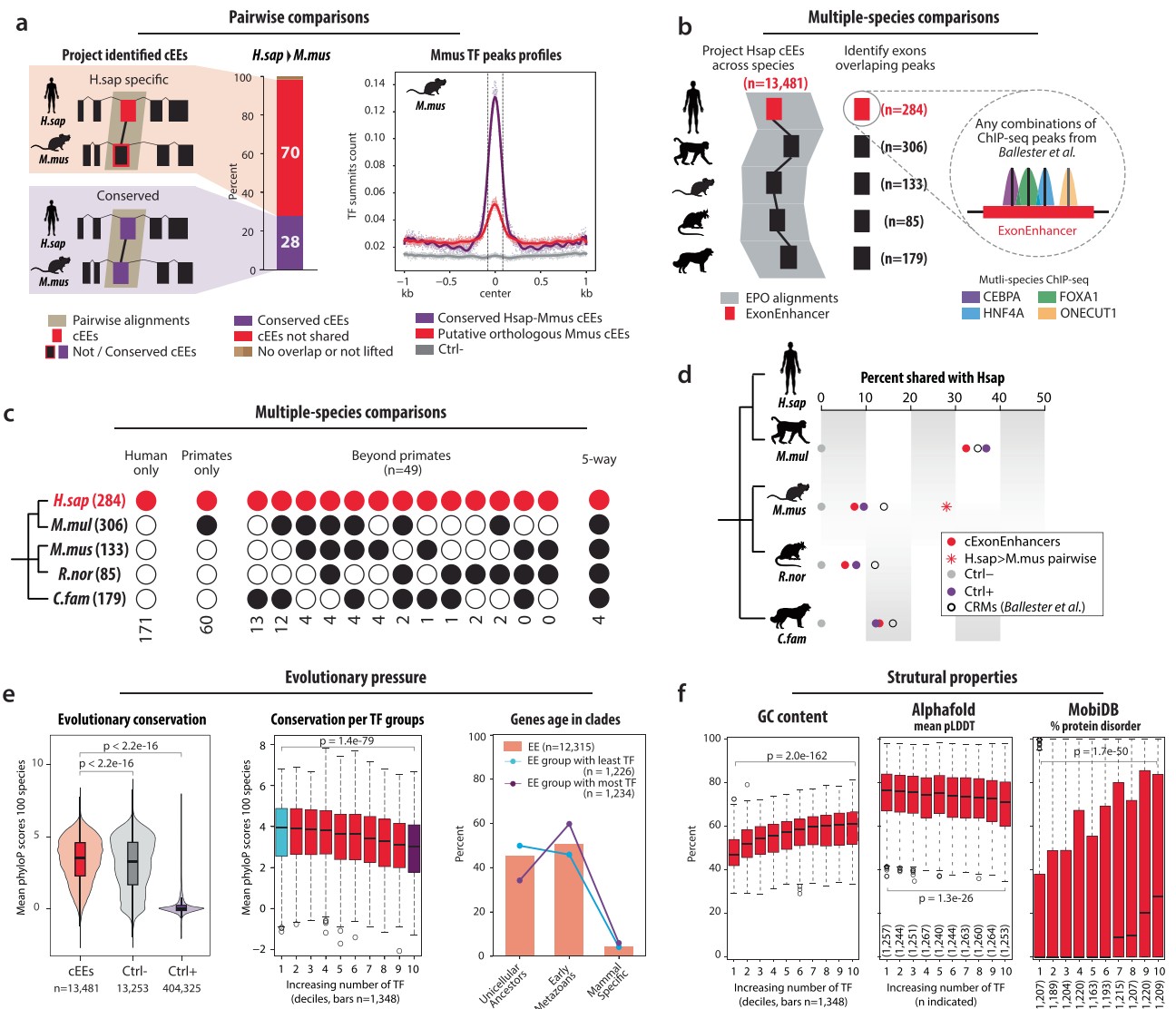

**Fig. 7 | Evolutionary conservation and selective pressure on exonic enhancers across mammals and beyond. a** Pairwise comparison of cEEs between human (*H. sapiens*, *H.sap*) and mouse (*M. musculus*, *M.mus*) using Ensembl pairwise genome alignments. *H.sap*-specific cEEs (red) are those not conserved in the mouse genome, while conserved cEEs (purple) meet the TF-binding criteria in both species. The bar plots quantify the fraction of *H.sap* cEEs that either map to a corresponding *M.mus* cEEs or do not, and the TF summit profiles for putative orthologous cEEs in *M.mus* (red line) indicate potential enhancer-like features even in cases lacking formal cEE annotation. **b** Schematic of a multi-species analysis leveraging data from Ballester et al., where four TFs (CEBPA, FOXA1, HNF4A, and ONECUT1) were profiled by ChIP-seq in five mammals (human, macaque, mouse, rat, and dog). Exonic regions exhibiting TF binding in each species were identified through multiple-genome alignments (EPO), permitting direct comparisons to *H.sap* cEEs. **c** Multi-species cEE occupancy analysis. Filled circles denote the presence of TF binding in that species. Black circles represent at least one TF peak, while red circles highlight the set of 113 human cEEs that retain exonic enhancer features across multiple species. **d** Proportion of human exonic enhancers (cEEs) shared with macaque (*M.mul*), mouse (*M.mus*), rat (*R.nor*), and dog (*C.fam*). Red circles indicate the percentage of cEEs validated across multiple species analyses, while the red asterisk highlights the subset of cEEs specifically conserved between human and mouse (panel a). Purple, grey, and open circles represent classic enhancers (Ctrl + ), negative controls (Ctrl-), and cross-species conserved regulatory modules (CRMs) from Ballester *et al.*, respectively. **e** Evolutionary constraints on cEE sequences. Box

plots (left) show mean phyloP scores across 100-species alignments, revealing stronger conservation for cEEs (*n* = 13,481) than for classic enhancers (Ctrl + , *n* = 404,325; or Ctrl-, *n* = 13,253). Each data point represents one cEE or one exon for controls. Boxes show the median and interquartile range; whiskers extend to 1.5xIQR. Paired two-sided Kruskal–Wallis tests with Bonferroni correction were used. *P* values are shown on the plot. (Middle plot) Further subdivision of cEEs by increasing TF occupancy indicates that lowly bound cEEs exhibit higher sequence constraint, while cEEs with high TF occupancy display relatively lower constraint (each bar, *n* = 1348, data point represents one cEE). (Right) Ancestral gene age analysis suggests that low-TF cEEs are more frequently associated with ancient gene families (unicellular ancestors, *n* = 5571), whereas high-TF cEEs are often linked to early metazoan genes (*n* = 6223, mammals specific n = 521). Data point represents one cEE. **f** Structural properties of cEEs grouped by TF occupancy (deciles, x-axis, each bar, *n* = 1348 unless otherwise specified in brackets, data point represents one cEE. Left: GC content (percentage) increases with higher TF occupancy. Middle: Mean pLDDT scores from AlphaFold predictions decrease with increasing TF occupancy, indicating lower confidence in protein structure prediction. Right: Percentage of cEEs overlapping with predicted disordered protein regions (MobiDB) increases with TF occupancy. Box plots show median (middle line), first and third quartiles (box), and 1.5x interquartile range (whiskers); outliers shown as individual points. Statistical significance was assessed using two-sided Kruskal–Wallis tests with Bonferroni correction. Source data are provided as a Source Data file and Zenodo.

recently evolved metazoan genes (Fig. 7e, middle and right, Supplementary Fig. 21).

To investigate the structural and biochemical properties of cEEs, we analysed how GC content, 3D protein structure stability, and intrinsic disorder change with increasing TF occupancy across the four species (Fig. 7f, and Supplementary Fig. 30). Grouping cEEs into deciles of TF binding density revealed a positive correlation between TF occupancy and GC content, mirroring the reduced sequence conservation in highly bound cEEs (Fig. 7e). In contrast, mean pLDDT scores, AlphaFold's confidence metric for predicted 3D structure[36], declined steadily with increasing TF occupancy. A second, orthogonal predictor of disorder (MobiDB[37]) showed the same trend: the fraction of cEE residues classified as intrinsically disordered rose with TF occupancy. Both metrics indicate that cEEs that recruit many different TF are preferentially situated in intrinsically disordered, and therefore structurally flexible, protein regions. Because such regions tolerate amino-acid variation without disrupting protein folding, they provide an evolutionary "safe harbour" where nucleotide-level changes that create additional TF-binding motifs impose minimal structural cost on the protein.

### What makes candidate exonic enhancers noteworthy?

Taken together, our analysis highlights several features that distinguish cEEs at genome scale. Notably, cEEs are strongly enriched for the poised/active enhancer mark H3K4me2 (7.4-fold over expectation; Supplementary Fig. 12), a signature that, to our knowledge, has not been quantified previously in the exonic-enhancer context. A minority of coding exons meet stringent enhancer criteria, yet cEEs are distributed across many genes. Despite harbouring fewer TF motifs than canonical intergenic enhancers (Fig. 2e; Supplementary Fig. 14), cEEs achieve comparable reporter activity (Fig. 4b), indicating efficient use of sequence under dual constraint. Population-genetic metrics indicate stronger contemporary purifying selection than in control exons (lower allele frequencies, higher synonymous/total ratio, lower pN/pS; Supplementary Table 2; Supplementary Fig. 16), while long-term patterns (higher dN/dS, positive DoS and FI > 1) point to episodic, compensatory amino-acid changes. Finally, cEEs with the heaviest TF load preferentially fall in intrinsically disordered protein segments (Fig. 7f), suggesting that proteins accommodate embedded regulatory motifs in structurally flexible 'safe harbours. Locating cEEs in such segments provides evolutionary "room": the exon can acquire nucleotide-level motifs that recruit multiple TFs while the resulting amino-acid changes have minimal structural impact on the protein.

## Discussion

Our genome-wide survey shows that exonic enhancers, although recognised at individual loci, represent a quantitatively substantial and functionally influential layer of regulation. In this study, we uncovered a widespread occurrence of cEEs across multiple species, using large-scale TF-binding maps, open chromatin profiles, and enhancer-reporter assays. Our results indicate that, while many exons merely encode protein domains, a substantial subset also harbours hallmark enhancer features; DNase hypersensitivity, histone modifications, G-quadruplex formation, and TF co-occupancy; underscoring their capacity to regulate gene expression. Previous studies documented exonic enhancer activity at isolated loci; our work extends this concept genome-wide, adds functional and evolutionary analyses, and delineates novelty versus confirmation.

Intronic and exonic DNA share the same chemical substrate, but coding exons carry additional informational load (reading frame, codon usage, and, for structured domains, side-chain constraints). Demonstrating that thousands of exons can accommodate dense TF-binding while preserving coding requirements, and quantifying the evolutionary and functional trade-offs that enable this, extends

beyond documenting their existence or novelty. Our analysis reveals how frequently the genome resolves this conflict, how strong the resulting enhancer outputs are, where within proteins such sequences reside, and which selective forces shape them.

Beyond their prevalence, cEEs can influence key biological processes. Functional assays, including STARR-seq and luciferase reporter assays, confirmed that cEEs drive transcription in cis. Notably, these assays measured enhancer activity solely from the exonic sequences, demonstrating that the exon itself, and not its intronic flanking regions, harbours cis-regulatory potential. Although 'strong' elements constitute a small fraction in both libraries, as is typical for STARR-seq, exon enhancers reach strong reporter outputs at rates comparable to, or slightly exceeding, intergenic enhancers, despite tighter sequence constraints. Because we used exon-bounded fragments (no intronic sequence) and the same minimal promoter and thresholds across libraries, this comparison reflects intrinsic enhancer capacity rather than library design artifacts. Furthermore, promoter capture Hi-C and ENCODE-rE2G data reveal that many cEEs physically interact with gene promoters, consistent with classical enhancer looping. In agreement, CRISPRi experiments targeting selected cEEs confirmed their regulatory function on both host and distal neighbouring genes. Mechanistically, these EE elements may integrate multiple TF inputs within coding regions, effectively bridging the coding and regulatory layers to coordinate gene expression.

Our investigation also highlights the potential for cEEs to shape disease phenotypes. By intersecting cEEs with genomic variation datasets (e.g., gnomAD, TCGA), we identified both common and rare variants that overlap critical TF-binding sites. Consistent with genome-wide constraint, cEEs carry an excess of ultra-rare variants, and although variant counts increase with TF load, the allele-frequency spectrum is nearly unchanged across TF deciles. These mutations, ranging from synonymous to nonsynonymous, can markedly alter cEE-driven enhancer activity in reporter assays, suggesting that even "silent" coding changes may have regulatory consequences. Moreover, exons with weak or no enhancer activity could acquire regulatory functions through the incorporation of novel variants. In cancer, cEE variants were associated with differential expression of putative target genes, which in turn correlated with patient survival. These findings expand emerging evidence that coding-region mutations can exert non-canonical effects on gene regulation, thereby influencing cancer progression and other disease processes[12].

From an evolutionary perspective, cEEs display a complex pattern of conservation and innovation across vertebrate lineages. Approximately 28% of human cEEs show conserved regulatory signatures in mouse, while others could exhibit lineage-specific patterns, suggesting dynamic regulatory evolution. The elevated sequence constraint observed in cEEs, compared to typical coding exons or classical enhancers, indicates strong purifying selection to maintain both coding and regulatory functions. Moreover, the correlation between TF occupancy and gene age suggests distinct evolutionary trajectories: ancient genes tend to harbour cEEs with lower TF complexity, whereas metazoan-specific genes often contain cEEs bound by multiple factors. It remains an open question whether coding exons acquire enhancer-like activity through variant acquisition, or whether pre-existing intronic enhancers accumulate mutations that eventually incorporate them into coding sequences.

As implied before[12], these findings confirm the necessity of a fundamental change in interpreting genetic variation in coding regions. Due to the dual functionality of cEEs, coding-sequence mutations should be evaluated not only for their impact on protein structure but also for potential regulatory consequences. This perspective has immediate implications for variant-interpretation pipelines, which must extend to coding exons to detect potential enhancer disruptions in research and clinical settings, particularly in cancer genomics where coding mutations are routinely assessed for

pathogenicity. The comprehensive cEE catalogue provided here enables such dual-impact assessments feasible.

Looking ahead, investigating the tissue-specific or cell type-specific contexts in which cEEs operate will be critical to understanding their broader biological impact. Future efforts should also extend cross-species comparisons to reveal the evolutionary forces shaping cEEs over longer timescales. By highlighting their prevalence, functionality, and pathological significance, our genome-wide analysis confirms and broadens the growing recognition that exons are not just blueprints for protein, but also an integral part of the regulatory machinery driving precise spatiotemporal gene expression.

## Methods

### Genomic annotation resources

Annotations for coding exons and transcripts were obtained from the UCSC Table Browser[35] using GENCODE[38] v41 (hg38) for human (*Homo sapiens*), GENCODE M33 (mm39) for mouse (*Mus musculus*), Ensembl[39] Release 104 (BDGP6/dm6) for fruit fly (*Drosophila melanogaster*), and Ensembl Release 44 (TAIR10) for *Arabidopsis thaliana*. For Transcription Start Site (TSS) data, FANTOM5[17] TSS peaks were used for human and mouse, with mm10 TSS peaks converted to mm39 using UCSC LiftOver. For *Drosophila melanogaster*, TSS peaks were obtained from ModENCODE[40] CAGE-seq experiments (NCBI accession number SRP001602) using the GEP UCSC Table Browser. For *Arabidopsis thaliana*, a comprehensive TAIR10 TSS catalogue was generated by integrating results from these studies[41-43]. Transcriptional regulators binding catalogues were obtained from ReMap[13] (https://remap.univ-amu.fr/, version 2022), which provides a curated and integrated catalogue of TF ChIP-seq data across multiple species.

### Defining and characterising exonic enhancers

To systematically identify candidate Exonic Enhancers (cEEs), we first generated a comprehensive set of protein-coding exons using GENCODE v41 (*H. sapiens*), GENCODE vM33 (*M. musculus*), FlyBase r6.54 (*D. melanogaster)*, and Araport11 (*A. thaliana*) annotations. Overlapping isoform exons with a size difference of ≤50 bp were merged to avoid fragmenting potential regulatory elements. To minimise confounding signals from promoter-proximal regions, we removed exons within 1 kb (human, mouse) or 100 bp (fly, Arabidopsis) of annotated transcription start sites (TSS) or termination sites (TES) in the same orientation. We further filtered out exons overlapping TSS peaks within 500 bp (human, mouse) or 100 bp (fly, Arabidopsis) windows. EE candidates were defined using transcription factor ChIP-seq peaks from ReMap. Candidate selection required: (i) a minimum of 10 distinct TFs overlapping the exon, (ii) the maximum TF density within the coding exon must exceed the local TF density measured in the ±50 bp flanking exonic region, to avoid artificially capturing TF clusters near exon-intron boundaries, and (iii) a minimum ratio of 0.5 between TF peak summits and total TF peaks overlapping the exon (Supplementary Fig. 31). For human, additional filtering was applied to refine EE candidates. Only exons from transcripts with a transcript support level (TSL) of 1 were retained, ensuring high-confidence annotations. All cEEs are provided in the Supplementary Data 4, also present on *Zenodo* (https://doi.org/10.5281/zenodo.17208730).

Furthermore, for human and mouse, exons overlapping promoter-associated chromatin marks from ENCODE cCREs[14] were removed. In the case of the mouse genome, mm10 promoter marks were lifted to mm39 using UCSC LiftOver to maintain consistency across genome builds.

For control datasets, negative controls were defined as protein-coding exons that passed the same filtering process as cEEs but had no overlapping TFs (Human n = 13,253; Mouse n = 18,457; Fly n = 903; Arabidopsis n = 7,862). Positive controls consisted of intergenic distal enhancers from ENCODE cCREs[6] (Human n = 404,325; Mouse n = 149,241) or the literature (Fly n = 133,253; Arabidopsis n = 9,025)

that passed the same filtering process as cEEs and overlapped at least 10 TFs, ensuring that control elements were subject to equivalent selection criteria.

### Permutation-based enrichment tests

For each set of genomic intervals (e.g., DHS peaks, TF-ChIP peaks, H3K27ac/H3K4me1 peaks) we quantified overlap with coding exons when analysing DHS or ReMap TF-ChIP datasets, and with cEEs for all other features by comparison to 10,000 sequence-matched random sets. Every real interval was shuffled once per permutation to a location on the same chromosome that (i) matched its length exactly and (ii) fell within its GC content bin, using a uniquely mappable genome mask. The fraction of shuffled intervals intersecting the test region defined the null distribution; fold-enrichment was calculated as observed/mean-expected and empirical two-tailed p-values as $2 \times (k+1)/(10\,000+1)$, where k is the number of permutations whose test statistic was at least as extreme as the observed value (i.e. greater than or equal to the observed if it lay above the null mean, or less than or equal if it lay below). Note on compact genomes: In *D. melanogaster* and *A. thaliana*, regulatory elements cover a large fraction of the genome, the sheer number of regulatory intervals relative to the limited genome size leaves little residual space for shuffling; this reduces the dynamic range of the permutation test, and results for these species should therefore be interpreted with caution. For TFBS enrichments within cEEs, permutations were performed similarly to genomic intervals but restricted to coding exons: for each permutation, motif instances within the coding regions universe were randomly reassigned to positions within coding exons. The fraction intersecting cEEs defined the null; fold-enrichment and two-tailed empirical p-values were computed as above (Supplementary Table 1). The relatively modest (yet significant) enrichment observed in the TFBS randomisation test can be partly attributed to the larger size of cEEs. As shown in Supplementary Fig. 14, cEEs are significantly longer than other coding exons, which naturally increases the number of elements that can be shuffled within them, as noted in the *bedtools shuffle* documentation.

### DNase-seq and ATAC-seq data processing

We compiled a comprehensive atlas of chromatin accessibility data from multiple sources. For human (GRCh38), DNase-seq peaks were obtained from ENCODE through the UCSC Genome Browser database. Mouse DNase-seq data (ENCODE accession ENCFF910SRW) was lifted from mm10 to mm39 using UCSC liftOver (minMatch=0.95). For both human and mouse, as well as *D. melanogaster* (dm6), we retrieved DNase-seq and ATAC-seq peaks with a significance threshold of 50 from ChIP-Atlas[8]. For *A. thaliana* (TAIR10), DNase-seq data was obtained from PlantRegMap[44]. ATAC-seq data was collected from the Gene Expression Omnibus under accessions GSE101482, GSE101940, GSE122772, GSE164159, and GSE173834. Processed datasets for the four species are provided in a Supplementary Data 1 Omics. All peak coordinates were standardised to their respective reference genome builds: GRCh38 (*H. sapiens*), GRCm39 (*M. musculus*), dm6 (*D. melanogaster*), and TAIR10 (*A. thaliana*).

### STARR-seq atlas processing and analysis

We constructed a comprehensive STARR-seq atlas by curating 112 published datasets, and by compiling active (log2FC > 0) and inactive (log2FC ≤ 0) peaks across four species (*H. sapiens, M. musculus, D. melanogaster*, and *A. thaliana*). All STARR-seq peaks were standardised to ENCODE narrowPeak format. Peak coordinates from earlier genome builds were converted to current assemblies (GRCh38, GRCm39, dm6, TAIR10) using UCSC liftOver (minMatch=0.95). Statistical significance was assessed using reported q-values or p-values where available (p < 0.05). A complete list of datasets and their sources is provided in Supplementary Data 2 STARR-seq catalogues. The biotype signatures of cEEs were determined by first assigning each ChIP-seq cell line to a

biotype as previously described[45], followed by calculating the normalised abundance of each biotype across cEEs. For K-562 and A-549 STARR-seq analysis, we generated a composite signal track using deepTools[46] v3.5.1. Specifically, fold-change signals from replicates were computed using bigwigCompare, then merged using bigWigMerge. Used datasets for the signal tracks are also listed in the Supplementary Data 2 STARR-seq catalogues.

## G-quadruplex analysis

Potential G-quadruplex (G4) forming sequences were identified using G4-seq data generated by an optimised G4 sequencing method[47]. The G4 peak data were retrieved from the Gene Expression Omnibus (GEO) under accession number GSE110582. The enrichment of G4 structures in cEEs compared to control regions (Ctrl+ and Ctrl-) was assessed using Fisher's exact test (one-sided, p < 0.05).

## TF peaks and TF binding site matching and randomisation

For each cEE, we quantified the overlap ("correspondence rate") between transcription factor (TF) fragments (score ≥ median) intersecting the cEE and their corresponding transcription factor binding sites (TFBSs) derived from the JASPAR2022 database[29] (https://jaspar.elixir.no/). These TF peaks were then randomly redistributed across the genome, while preserving the same chromosome assignment and GC-content bin, to generate a null distribution of correspondence rates. The observed correspondence rate in cEEs was subsequently compared with the rate obtained under randomisation.

## GTEx PSI analysis

Percent-spliced-in (PSI) values were obtained from the multi-tissue splice-junction dataset of Lappalainen et al[48]., which reports exon-skipping PSI for 18 GTEx tissues selected for high sample depth and broad gene coverage. For each tissue we calculated, for every cEE, the mean PSI across all samples (excluding NA values). The resulting per-cEE PSI values were used to generate tissue-specific distributions (boxplots, Supplementary Fig. 9); we also report the tissue-level median and mean PSI across all cEEs.

Isoform status was assessed with APPRIS[49] v50. Human, mouse and fly coding exons were intersected with APPRIS transcripts, and each exon was labelled as PRINCIPAL (M, 1 or 2), PRINCIPAL (3-5), ALTERNATIVE (M, 1 or 2), MINOR or UNKNOWN. We then computed the proportion of cEEs and Ctrl- exons falling into the three broad classes used by APPRIS (principal isoforms (PM/P1/P2), other principal, and non-principal) separately for each species (Supplementary Fig. 10). Differences between cEEs and controls were tested with $\chi^2$, and effect size was quantified with Cramér's V. Although $\chi^2$ p-values were <$10^{-9}$ in all three species, Cramér's V was only 0.05-0.07, indicating that the distributions differ statistically but not materially.

## GTEx tissue-expression analysis for cEE host genes

Note that GTEx analyses reflect expression of cEE host (or predicted target) genes and do not measure enhancer activity per se. cEEs were first assigned a tissue "biotype signature" based on the predominant cell-type label of the ReMap ChIP-seq peaks that support each enhancer (for example, a cEE bound in liver ChIP-seq experiments was labelled Liver), whereas one cEE bound in lung epithelium was labelled Respiratory. Gene-level TPM matrices from GTEx[50] v8 were then downloaded and consolidated into 18 major organ classes by merging closely related sub-tissues (for example, Brain - amygdala and Brain - cortex were combined into Brain/Nervous). For every gene the TPM values were min-max normalised across these 18 tissues to permit comparison on a common scale.

To visualise tissue breadth, we generated violin plots for each biotype group (Supplementary Fig. 18). For every cEE in the group, we recorded the TPM of its host/target gene in three contexts: the "matching" tissue that corresponds to the cEE's own biotype and two randomly selected, unrelated tissues. The distributions of these three sets were compared with two-sided Mann-Whitney tests; significance thresholds were set at $p < 0.05$.

For a complementary overview we produced a heatmap (Supplementary Fig. 19). Within each biotype group we calculated the median normalised TPM of all host genes across the 18 tissues and converted this vector to Z-scores, thereby emphasising relative over absolute expression. Rows were ordered in two steps: first by the rank of the highest Z-score in the matching tissue (groups whose peak expression occurs in their own tissue appear at the top) and second, for ties, by the magnitude of that peak Z-score. Thus Reproductive_M cEEs, whose median expression is highest in male tissues (e.g., testis); occupy the first rows, followed by Reproductive_F, Endocrine, and other biotypes in descending order of tissue-specific prominence.

## Uhlén-based tissue specificity classification

Each cEE was assigned a tissue "biotype signature" based on the predominant tissue or cell-type annotation of its associated ReMap ChIP-seq peaks. We then retrieved tissue expression classifications for the host and target genes of these cEEs using the dataset from Uhlén et al[51]. which integrates RNA-seq data from the Human Protein Atlas (HPA) and GTEx. In this classification system, genes are labelled as "tissue enriched," "group enriched," "tissue enhanced," "low tissue specificity," or "not detected." For analysis, we collapsed these into three categories: low specificity ("low tissue specificity" or "not detected"), elevated expression in unrelated tissues ("tissue enriched," "group enriched," or "tissue enhanced" in tissues other than the cEE biotype), and biotype-matched elevated expression ("tissue enriched," "group enriched," or "tissue enhanced" in the same tissue as the cEE's biotype signature).

For the horizontal bar plot (Supplementary Fig. 20), we calculated the proportion of host and target genes in each biotype group falling into these three categories. The resulting plot visualises how often cEEs are associated with genes showing elevated expression in their "matching" tissue compared to unrelated tissues or those broadly expressed.

To assess whether the distribution of expression categories differed significantly across gene sets, we generated a stacked bar plot (Supplementary Fig. 20) including all cEE-associated genes (host and target), negative controls, and all protein-coding genes. A Chi-squared test indicated a highly significant difference in tissue specificity category distribution across these groups ($p$ value = $2 \times 10^{-66}$), although the effect size, measured by Cramér's V, was modest (V = 0.067).

We also calculated the tissue-specificity index $\tau$ for each gene using the RNA-seq matrix from Uhlén et al[52]. which quantifies expression breadth across tissues. $\tau$ values range from 0 for ubiquitously expressed genes to 1 for highly tissue-specific genes. Violin plots were generated for the $\tau$ distributions of all genes, cEE host and target genes, and control genes (Supplementary Fig. 20 d). Statistical comparisons were performed using two-sided Student's t-tests, with significance set at p < 0.05.

## ENCODE RNA-seq analysis of cEE-associated gene expression across cell lines

To assess the expression profiles of cEE-associated genes, we used RNA-seq data from the Expression Atlas dataset entitled "RNA-seq of long polyadenylated and long non-polyadenylated RNA from ENCODE cell lines" (Supplementary Fig. 20e,f). From this dataset, we extracted FPKM values corresponding to the "whole cell, long polyA RNA" category for three ENCODE cell lines: K-562, A-549, and GM12878.

We focused on the STARR-seq subset of cEEs classified as "strong" (n = 61) based on their activity in the K-562 STARR-seq screen. For these cEEs, we retrieved the expression levels (FPKM) of their host genes and, where available, their predicted target genes in each of K-562, A-549, and GM12878 cell lines. Distributions of expression values

were compared using Wilcoxon signed-rank tests. Median expression was highest in K-562, with pairwise comparisons showing only a trend toward significance (lowest $p = 1.308 \times 10^{-2}$ for K-562 vs A-549; $p = 0.206$ for K-562 vs GM12878).

### Selection of candidate EEs for luciferase assays

cEEs for luciferase assays were selected based on the presence of K-562 STARR-seq and DNase-seq peaks. Single-nucleotide polymorphism (SNP) data in the hg38 reference genome were obtained from gnomAD[28] v3 (https://gnomad.broadinstitute.org/), and their potential impact on transcription factor (TF) binding was predicted using the FABIAN-variant tool[53]. Where necessary, synonymous SNPs were manually introduced for construct generation. Missense and synonymous variants tested for enhancer activity gain or loss were chosen based on their predicted effect on existing TFs and TFBSs within each candidate EE.

### Cell culture

The K-562 cell line (ATCC CCL-243), derived from chronic myelogenous leukaemia, was obtained from the American Type Culture Collection (ATCC) and cultured in RPMI 1640 medium with GlutaMAX (Thermo Fisher Scientific), supplemented with 10% foetal bovine serum (FBS; Thermo Fisher Scientific). Cells were incubated at 37 °C in a humidified atmosphere containing 5% $CO_2$, passaged every three days at a density of $2 \times 10^5$ cells/mL, and routinely tested for mycoplasma contamination.

### Experimental validation of exonic enhancer activity

We tested 23 sequences corresponding to identified exonic enhancers (Supplementary Table 4-6). Each sequence (lucEE1 to lucEE23) was cloned into the pGL3-Promoter vector (Promega, E1761), containing the SV40 promoter upstream of the luciferase gene, at the BamHI restriction site. For transfection, $1 \times 10^6$ K-562 cells were electroporated with 1 μg of each lucEE construct and 200 ng of Renilla control vector using the Neon Transfection System (Thermo Fisher Scientific; pulse voltage: 1450 V, pulse width: 10 ms, pulse number: 3) and cultured in 1 mL of medium in 24-well plates. After 24 h, cells were lysed, and luciferase activity was measured using the Dual-Luciferase Reporter Assay (Promega). Firefly luciferase data were normalised to Renilla luciferase activity, and results were expressed as fold-change in relative light units compared to the empty pGL3-Promoter vector. The number of biological replicates is indicated in the figure legends. Generated plasmids, reporter constructs, and CRISPRi guide sequences generated in this study are available from the corresponding author.

### Exonic enhancer activity gain or loss assays

To assess the impact of sequence modifications on enhancer activity, the same protocol described above was applied, using synthetic exon DNA fragments containing designed sequence-specific modifications (Supplementary Table 6). Luciferase activity was measured to determine whether these alterations led to gain or loss of function in enhancer activity.

### Luciferase constructs for promoter-enhancer evaluation

To test promoter activity, DNA fragments corresponding to the promoter regions of the *GARRE1* gene and its partner genes (*GPI* and *PDCD2L*), or the *USP20* gene and its associated partners (*C9orf78* and *TOR1B*), were synthesised and cloned into the MluI and XhoI sites upstream of the luciferase coding sequence, generating six reporter constructs (Supplementary Table 5). For each promoter construct, the corresponding wild type exonic enhancer (EE3_WT or EE19_WT) was cloned into the BamHI restriction site (Supplementary Table 4). Luciferase assays were performed following the same protocol described above for exonic enhancer activity assays. For Fig. 3b, c, effects were

large and consistent across experiments, and additional biological replication was not expected to alter the conclusions.

### EE STARR-seq processing and analysis

**Selection Criteria.** cEE tested in the STARR-seq experiments were primarily identified based on signatures derived from the K-562 cell line, complemented by additional cEEs exhibiting signatures in GM12878 and A-549 cells. Selection criteria included: (i) at least one non-redundant TF peak summit overlapping a DNase-seq peak, and (ii) at least one non-redundant TF peak summit coinciding with at least three enhancer peaks identified in the STARR-seq catalogue. Only cEEs ≥130 bp were retained. For cEEs exceeding 220 bp, the sequence was trimmed to a 220 bp region centred on the highest concentration of TF peaks, remaining within the exon boundaries (Supplementary Fig. 32a). For each cEE, we synthesised only the annotated coding-exon sequence, trimming precisely at exon boundaries and excluding all flanking intronic DNA, before cloning the fragment into the STARR-seq vector. This exon-only design prevents intronic enhancers, which can be adjacent to coding regions, from confounding interpretation of the reporter signal.

**SNP-modified sequences.** For K-562, single-SNP mutant constructs were generated by introducing synonymous variants selected from gnomAD v3 (https://gnomad.broadinstitute.org/) that were predicted to alter TF/TFBS binding by more than |0.66|, as estimated by the FABIAN-variant tool. Multi-SNP mutant constructs were generated in the same manner, but included at least two additional non-adjacent SNPs per sequence.

**Controls.** Negative controls were defined as regions lacking any ATAC-seq or DNase-seq peaks and exhibiting the lowest JASPAR2022 TFBS scores. Positive controls were randomly drawn from regions containing at least three TF peaks in each of the three cell lines. All control fragments were 130-220 bp in length. Additionally, 200 negative and positive "STARR + /-" technical controls were included based on published STARR-seq experiments[30]. The complete list of all 11,999 tested sequences is provided in the Supplementary Data 3 STARR-seq.

### STARR-seq library construction and transfection

A total of 11,999 candidate sequences were synthesised by Twist Bioscience, each appended with 5′-ACGCTCTTCCGATCT and 3′-AGATCGGAAGAGCAC, which are 15 bp invariable adaptors corresponding to TruSeq read indices. The resulting oligo library was resuspended in $H_2O$ at 10 ng/μL. To add homology arms for subsequent cloning (PCR_A), 10 ng of the oligo library was amplified in three parallel 50 μL reactions using KAPA HiFi Hotstart Readymix (Roche, cat. #07958927001), 10 μM forward primer Fw_Enh_MPRA (CAACTGATCTAGAGCATGCAACGCTCTTCCGATCT), 10 μM reverse primer Rv_Enh_MPRA (GAAGCGGCCGGCCGAATTCGTGTGCTCTTCCGATCT), and the following programme: 98 °C for 2 min; 20 cycles of (98 °C for 20 s, 60 °C for 30 s, 72 °C for 30 s); and a final extension at 72 °C for 2 min. The three PCR products were purified with Nucleospin cleanup (Macherey-Nagel, cat. #740609.50) in a 20 μL elution, then desalted on a MF-Millipore MCE 0.025 μm membrane (Sigma, cat. #VSWP02500) for 25 min.

The STARR-seq vector was digested with AgeI-HF, SalI-HF, and NotI-HF (NEB, cat. #R3552S, #R3138S, #R3189S), loaded onto a 1.5% agarose gel at 100 V for 45 min, and the desired band was extracted using the Nucleospin Gel Cleanup kit (Macherey-Nagel, cat. #740609.50). For Gibson Assembly (NEB, cat. #E5510S), 150 ng of digested vector and 40 ng of the purified PCR_A product were incubated in two separate 20 μL reactions at 50 °C for 45 min. The assembled products were then desalted on a MF-Millipore MCE 0.025 μm membrane for 25 min.

**Transformation and Library Expansion.** Each 15 μL Gibson assembly reaction was electroporated into 40 μL of MegaX DH10B T1R cells (Thermo Fisher, cat. #C640003) using a Bio-Rad Gene Pulser II system (2 kV, 200 Ω, 25 μF) in 0.1 cm cuvettes. Cells were recovered for 1 h at 37 °C in 8 mL SOC medium. To estimate library coverage, 1 μL of the recovery culture was plated on LB + carbenicillin (Invitrogen, cat. #10177-012) and grown overnight at 37 °C, yielding ~775 colonies per μL (516x coverage for 12,000 inserts). The remaining transformation was spread over eight 15 cm LB + carbenicillin plates and grown for 16 h at 37 °C. Colonies were then transferred to 1 L of Terrific Broth with 1x carbenicillin and incubated at 37 °C, 100 rpm for 16 h. Ten midipreps (Nucleobond Midiprep Kit, Macherey-Nagel, cat. #740410) were combined, eluted in a total of 500 μL, and yielded 1,751 μg of Exonhancer plasmid library DNA.

**Cell Culture and Transfections.** K-562 cells were maintained in RPMI 1640 (Thermo Fisher) and passed for at least one week prior to transfection, with 97% viability at the time of use (passage +10). For transfection, $50 \times 10^6$ cells were pelleted, washed in PBS, and resuspended in 1 mL Neon resuspension buffer (Thermo Fisher). A total of 250 μg Exonhancer plasmid library was added, and 10 transfections (100 μL each) were performed (settings: 1450 V, 10 ms, 3 pulses). Two transfections (200 μL total) were combined per replicate in a culture dish with 10 mL of RPMI 1640 and incubated at 37 °C for 24 h, creating five replicates. As a control, one replicate was transfected using the STARR-seq empty vector using the same settings. 24 h after transfection, GFP was assessed as an indication of enhancer activity (Supplementary Fig. 32b).

**gDNA Isolation and STARR-seq Protocol.** After 24 h of incubation, gDNA was extracted from 1 mL of each of three replicates using the Flexigene protocol in a final volume of 50 μL to assess input library integrity. These gDNA samples, plus a midiprep reference, were each PCR-amplified (10 ng input) using KAPA HiFi (50 μL reactions) with primers Fw #11 (GGGCCAGCTGTTGGGGTGAGTAC) and Rv #10 (CTTATCATGTCTGCTCGAAGC) under the following programme: 98 °C for 2 min; 15 cycles of (98 °C for 20 s, 65 °C for 20 s, 72 °C for 1 min); 72 °C for 2 min. Products (~1350 bp) were purified (Nucleospin) and quantified by dsDNA HS Qubit.

For the STARR-seq assay, total RNA was extracted (RNeasy, Qiagen), followed by mRNA isolation (Dynabeads, Invitrogen) for each of the 5 replicates. cDNA was synthesised using Superscript III (Thermo Fisher) with a reverse transcription primer (CAACTCATCAATGTATCTTATCATG) and an RNase H step at 37 °C for 20 min. cDNA was purified, quantified via ssDNA Qubit, and amplified (PCR1) using KAPA HiFi and primers Fw (GGGCCAGCTGTTGGGGTGTCCAC) and Rv (CTTATCATGTCTGCTCGAAGC) under the programme: 98 °C for 2 min; 15 cycles of (98 °C for 20 s, 65 °C for 20 s, 72 °C for 1 min); and 72 °C for 2 min. A second indexing PCR (dual-index TruSeq) was conducted (20 cycles) on PCR1 products from both gDNA and cDNA samples, followed by purification (AmpureXP) and verification of product size (240-360 bp) by Qubit and Bioanalyzer. Libraries were sequenced on a NextSeq 2000 (Illumina).

## STARR-seq processing

Sequencing paired-end reads were aligned to the reference library using Bowtie2 with the "very-sensitive" parameter set to ensure high-fidelity mapping. Resulting BAM files were processed with SAMtools[54] to remove unmapped reads and any reads aligned to the reverse strand (using samtools view -F 20). The filtered reads were then sorted and indexed, and unique identifiers in the library were quantified to produce raw read counts for each sample. Counts were normalised to counts per million (CPM) to enable cross-sample comparison. Finally, the pipeline computed the ratio and $\log_2$ ratio of normalised cDNA counts to the corresponding input sample. Sequences with an input read count <500 or with an input STD >= 20 were discarded (Supplementary Fig. 31b). SNP-associated effects on enhancer activity in K-562 cells were assessed according to Long et al. method[55] using a two-sided Wald test, with significance set at FDR < 0.05 (Benjamini-Hochberg correction).

## gnomAD-based constraint analysis of candidate exonic enhancers

Whole-genome single-nucleotide variants (SNVs) from gnomAD v3.12 (~76 000 unrelated individuals) were filtered for "PASS" status and overlapped with human cEEs and negative-control coding exons (Ctrl-) using *bedtools intersect*. For every exon we recorded the allele frequency (AF) of its SNVs and the fraction of SNVs that were ultra-rare (AF ≤ 0.01 %). Counts of synonymous and missense SNVs were normalised by exon length to obtain per-base densities. Potential synonymous (S) and nonsynonymous (N) sites were enumerated from the codon sequence, allowing calculation of the synonymous-to-total variant ratio and the population ratio pN/pS = (missense/N) ÷ (synonymous/S).

To assess longer-term divergence, each human cEE was aligned to its mouse orthologous exon using a LASTZ alignment[39]. Per-codon dN and dS were estimated using the Nei-Gojobori method[56] (dnds python package), yielding both per-exon and pooled dN/dS values. Combining polymorphism and divergence counts gave the Direction of Selection statistic DoS = dN/(dN+dS) - pN/(pN+pS) and the Fixation Index FI = (dN/dS) / (pN/pS). Per-exon distributions were compared with two-sided Mann-Whitney tests, whereas pooled counts (allele-frequency bins, pN/pS, dN/dS) were evaluated with $\chi^2$ tests; all results are summarised in Supplementary Table 2.

**Variant-density and TF-load analyses.** For germline (gnomAD) and somatic (TCGA Pan-Cancer) data, variant counts were normalised by exon length to obtain per-base densities (Supplementary Fig. 21). Densities were compared among cEEs, Ctrl- exons and intergenic enhancers (Ctrl + ) with two-sided Student's t-tests. To relate density to regulatory load, the number of distinct ReMap2022 transcription-factor summits within each cEE was counted and cEEs were ranked and divided into ten equal-sized deciles. Within-cEE density differences across deciles were assessed with Kruskal-Wallis tests.

**Allele-frequency spectrum across TF deciles.** All gnomAD SNVs in cEEs were assigned to four AF bins; ultra-rare (≤ 0.01 %), rare (0.01–0.1 %), low-frequency (0.1–1 %) and common (> 1 %); inside each TF decile (Supplementary Fig. 22). Stacked barplots display the proportional spectrum, and a companion plot shows the raw SNV counts. While the Chi-square test yielded a highly significant *p* value ($p = 5.76 \times 10^{-14}$), indicating that the null hypothesis of identical distributions across groups can be statistically rejected, the effect size (Cramér's V = 0.0068) is extremely small, indicating that, despite statistical significance, the actual differences in allele frequency category distributions across TF groups are negligible in practical or biological terms.

## Regression-to-the-mean simulation

To assess whether the observed trend (where synonymous variants tend to decrease enhancer activity in highly active elements and increase it in weak ones) could be explained by a regression to the mean, we implemented a simulation-based null model accounting for experimental noise. For each wild-type (WT) insert with at least three technical replicates, we calculated the mean $\log_2$-transformed enhancer activity ($\log_2$FC_WT) and its replicate standard deviation ($\sigma_i$), representing the expected noise. Observed variant effects were defined as ΔA = $\log_2$FC_variant - $\log_2$FC_WT, and regressed against $\log_2$FC_WT to estimate the empirical slope ($\beta$_obs) (Supplementary Fig. 23).

To generate the null distribution, we simulated 10,000 pseudo-variants by drawing a synthetic activity value A_sim from a normal distribution $N(\log_2FC\_WT, \sigma_i^2)$ for each WT insert. For each simulation, we calculated $\Delta A\_sim = A\_sim - \log_2FC\_WT$ and regressed it on $\log_2FC\_WT$, recording the slope ($\beta\_sim$) (Supplementary Fig. 23). The empirical p-value was defined as the proportion of simulations where $\beta\_sim \leq \beta\_obs$. This approach preserves the variance structure of each insert while testing the null hypothesis that observed variant effects arise solely from measurement noise.

## CRISPRi targeting of cEEs

### Generation of CRISPRi-competent K-562 cells.
A CRISPRi-competent K-562 clone was generated by transducing cells with a dCas9-KRAB-MeCP2 lentiviral vector (Addgene #122205). Lentiviruses were produced in HEK293T cells co-transfected with the VSVG packaging plasmid via calcium phosphate precipitation. K-562 cells ($0.3 \times 10^6$ cells/mL) were infected twice in the presence of blasticidin selection. Single cells were seeded into 96-well plates were subsequently infected with a Crop-seq-derived lentivirus (Addgene #86708) encoding an sgRNA targeting the human *CD81* promoter (Supplementary Table 7), followed by seven days of puromycin selection. Individual clones were assessed for CD81 surface expression by cytometry and validated by RT-qPCR. A clone exhibiting efficient CD81 knockdown was chosen for further CRISPRi experiments (Supplementary Fig. 33).

### CRISPRi targeting of EEs.
sgRNAs against selected cEEs were designed using the CRISPOR tool[57], synthesised, and cloned into the Crop-seq-guide-puro vector (Genecust, Luxembourg) via the BsmBI site. A no-target sgRNA was also cloned as a control. Lentiviral vectors were amplified in Endura E. coli cells (Lucigen), and lentiviral particles were produced as described above. For each cEE, $2 \times 10^4$ K-562 cells expressing dCas9-KRAB-MeCP2 were independently transduced at a multiplicity of infection (MOI) of 10 with 2 mL of complete RPMI medium supplemented with 10% FBS and 1x Penicillin-Streptomycin-Glutamine (Thermo Fisher). Ten days post-infection, cells were harvested for RNA extraction. The list of sgRNA is provided in Supplementary Table 7. For negative controls we used one sgRNAs targeting a coding exon lacking obvious enhancers signatures (such as TF binding, open chromatin, histone marks) from each of the tested cEE containing-gene (primers in Supplementary Table 8).

### Gene expression analysis.
For CRISPRi readouts, we measured stable transcripts by RT-qPCR with exonic primer pairs (not junction-spanning). Total RNA was DNase-treated prior to reverse transcription; primer sequences are listed in Supplementary Table 5. Total RNA was extracted using the RNeasy Plus Mini Kit (Qiagen) following the manufacturer's protocol. One microgram of RNA was reverse transcribed with SuperScript™ VILO™ Master Mix (Thermo Fisher Scientific, #11755250). qRT-PCR was performed using SYBR Green Master Mix (Thermo Fisher Scientific) on a QuantStudio™6 Flex (Thermo Fisher Scientific) instrument, with 1:10 diluted cDNA. Relative expression was analysed by the $2\char`^{\Delta\Delta CT}$ method, normalised to GAPDH expression. Each group was tested in three independent RNA and cDNA preparations, and the mean ± standard deviation was calculated relative to the no-target control. Primers used are listed in Supplementary Table 8.

## Integration of Hi-C, ENCODE-rE2G, and eQTL data for cEE interaction analysis

To identify putative enhancer-promoter interactions, we overlapped cEEs mapped to the hg38 reference genome with the promoter capture Hi-C dataset[26]. Given the resolution of PCHi-C (25 kb-2 Mb), these interactions suggest spatial proximity but do not exclude potential contributions from adjacent regulatory elements. We retained cases in which (i) an EE resided in a target fragment interacting with a complementary fragment (n = 9340), or (ii) an EE lay within a baited (promoter) fragment interacting with another baited fragment (n = 1637). cEEs were further overlapped with predicted regulatory interactions from the ENCODE-rE2G[27] model using bedtools[58] intersect (option *-f 0.50*). Additionally, we intersected cEEs with expression quantitative trait loci (eQTLs) from GTEx[50] v8. We defined an "internal interaction" as an EE that targets a gene located within its own gene body, whereas "external interactions" refer to cEEs targeting genes outside their host gene body. Finally, "robust interactions" were those for which at least two of the above datasets (promoter capture Hi-C, ENCODE-rE2G, eQTLs) provided support.

## Impact of exonic enhancer mutations in cancer: TCGA PanCanAtlas Analysis

### Mutation Data Processing.
Somatic mutation data from The Cancer Genome Atlas (TCGA) were retrieved from the publicly available multi-cancer mutation annotation format (MAF, m) file provided by the TCGA PanCanAtlas group[59]. To integrate these data with cEEs, genomic coordinates of cEEs were lifted over from hg38 to hg19 using UCSC LiftOver, followed by bedtools intersect to identify overlaps between cEEs and somatic mutations.

### Differential Expression Analysis of EE-Target Genes.
To assess the impact of EE mutations on gene expression, we analysed differential expression in genes with robust cEE-target interactions (defined as interactions supported by at least two datasets: promoter capture Hi-C, ENCODE-rE2G, or GTEx eQTLs). Differential expression analysis was performed using pyDESeq2[60] (v0.4.8), comparing two groups of patients: i) Mutant Group: Patients carrying mutations within a given cEE, ii) Wild-type Group: Patients with no mutations in the corresponding cEE. Deseq2 TCGA results corresponding to Fig. 5d are reported in the Supplementary Data 4.

A separate analysis was conducted where only patients carrying synonymous (silent) mutations within a given cEE were retained and compared against those without mutations. Differentially expressed genes were identified based on an absolute fold-change threshold of |FC| > 0.5 and a significance cutoff of padj <0.05 (Benjamini-Hochberg correction). All cEE mutation annotations and associated expression changes are provided in the Zenodo repository (https://doi.org/10.5281/zenodo.17208730) and in the Supplementary table.

## PanCan survival analyses

To assess the potential clinical impact of EE-mediated gene dysregulation, we selected differentially expressed cEE-target genes identified in four cancer types: lower-grade glioma (LGG), lung adenocarcinoma (LUAD), stomach adenocarcinoma (STAD), and skin cutaneous melanoma (SKCM). These genes were chosen based on their differential expression patterns in TCGA PanCanAtlas data (see "TCGA PanCanAtlas Analysis" section) and their classification as robust cEE targets (i.e., supported by at least two of the following datasets: promoter capture Hi-C, ENCODE-rE2G, or GTEx eQTLs, Supplementary Data 4). To evaluate the prognostic significance of cEE-target gene expression, we used the cSurvival[61] tool (v1.0.6). Kaplan-Meier survival curves were generated for high- and low-expression groups, stratified based on the median expression of each target gene across cancer patients. Log-rank tests were applied to determine statistical significance ($p < 0.05$). For each cancer type, we assessed the overall survival (OS) of patients grouped according to target gene expression levels. Hazard ratios (HRs) and confidence intervals (CIs) were computed using a Cox proportional hazards model, adjusting for relevant clinical covariates such as tumour stage and patient age when available.

## Variant effects predictions and visualization in exonic enhancers

To assess how single-nucleotide polymorphisms (SNPs) may alter transcription factor (TF) binding within cEEs, we used the FABIAN-

variant tool[53]. FABIAN-variant quantifies binding affinity changes by evaluating the presence or absence of TF motifs at SNP-affected sites. For each variant (TCGA, gnomAD, Supplementary Fig. 34) located within an cEE, we computed its predicted effect on associated TFBSs. Mutations that significantly altered TF-binding affinity (binding gain or loss) were recorded. To facilitate the interpretation of TF-binding alterations, we visualized the computed TF-binding impacts using lollipop plots. These plots were generated using track visualization tools in UCSC Genome Browser[35]. These visualizations provide a systematic overview of how mutations in exonic enhancers may alter TF-binding landscapes, with potential implications for gene regulation in cancer. The full lollipop track illustrating these TFBS alterations is available within the listing of UCSC public hubs (https://genome.ucsc.edu/cgi-bin/hgHubConnect) and also public sessions (https://genome.ucsc.edu/cgi-bin/hgPublicSessions) as "ExonEnhancers" (https://genome.ucsc.edu/s/Benoit%20Ballester/hg38_ExonEnhancers_gnomAD; https://genome.ucsc.edu/s/Benoit%20Ballester/hg19_ExonEnhancers_TCGA).

### Processing GWAS catalogue and 1000 genomes SNP data

GWAS variants from the NHGRI-EBI GWAS Catalogue[62] (v1.0.2) and common human SNPs (in VCF format) from the 1000 Genomes Project[63] were retrieved. GWAS variants without rsIDs or genomic coordinates were removed. Common SNPs were filtered in PLINK (v1.9) using the parameters --maf 0.01, --geno 0.05, and --hwe 1e-6. Next, common SNPs within 1 Mb of each GWAS lead SNP and showing high linkage disequilibrium ($r^2 > 0.8$) were identified with PLINK (-ld-window-kb 1000 -ld-window-r2 0.8). Each GWAS trait was then mapped to one of the parent trait categories ("Disease," "Biological process," "Measurement," "Phenotype," or "Other") based on the Experimental Factor Ontology (EFO). Finally, we normalised the number of SNPs in each category by the total number of elements in the overlapped dataset (cEE or controls).

### Conservation and Evolutionary Comparisons

Cross-Species Projection of cEEs was performed using both pairwise and multiple genome alignments. For the pairwise approach, cEEs defined in the human genome (hg38) were mapped to the mouse genome (mm39) through the Ensembl API with LASTZ-net alignments (Ensembl release 104). High-confidence mappings were obtained by retaining regions with at least 90% sequence identity over 50 bp, and coordinate conversion was conducted using a chain-based liftOver. For the multiple genome alignment approach, cEEs in hg38 were projected onto mm39 (*Mus musculus*), mml10 (*Macaca mulatta*), rnor7 (*Rattus norvegicus*), and cfam1 (*Canis lupus familiaris*) via the Enredo-Pecan-Ortheus[64] (EPO) alignments. Coordinate conversion blocks were retrieved from Ensembl, and in-house Perl scripts (v3.8) were used to identify orthologous exonic regions in each genome.

### Transcription factor ChIP-seq analysis across species

ChIP-seq data was retrieved from Ballester et al[34]., which profiled four key transcription factors (CEBPA, FOXA1, HNF4A, and ONECUT1) in the liver of human, macaque, mouse, rat, and dog. Reads were pooled prior to alignment and were trimmed and filtered using fastp (v.0.23.1) with the "--max_len1 36 --cut_tail --cut_tail_mean_quality 25 --average_qual 20" parameters. Reads were aligned to the respective genomes and outputted in BAM format using bowtie2 (v.2.5.1) and samtools view (v.1.18) under default parameters. Unmapped, duplicate, and multimapping reads were removed using sambamba (v.1.0.0). Peak calling was performed with MACS2 (v.2.2.7.1) callpeak, applying the effective genome sizes ("-g" parameter) computed with unique-kmers.py from the khmer[65] programme. The resulting TF peak sets were then intersected with the projected cEEs using BEDTools (v.2.30.0), and cross-species enhancer occupancy was assessed based on regions that overlapped at least one TF peak in each lineage.

### Computing evolutionary conservation scores

PhyloP conservation scores were obtained from multiple sources: human (hg38) 100-species phyloP scores, mouse (mm39) 35-species scores, and *Drosophila melanogaster* (dm6) 124-species scores were retrieved from the UCSC Genome Browser, while *Arabidopsis thaliana* (TAIR10) 63-species scores were acquired from PlantRegMap[34]. Mean phyloP scores for each cEE were calculated using bigWigAverageOverBed (Kent utilities suite, v2), excluding regions that lacked phyloP coverage. Comparisons of cEE conservation scores were made against control coding exons (without enhancer activity) and classic intergenic enhancers. The Kruskal-Wallis test ($p < 0.05$) was used to compare phyloP conservation among cEEs, focusing on the first and last deciles of each distribution.

### Gene age analysis

Gene age categories were retrieved from Trigos et al.[66]. cEEs were mapped to their host genes based on GENCODE (release 41) annotations, and each gene was classified according to its evolutionary origin. cEEs were stratified by gene age category (e.g., unicellular, metazoan), and enrichment analyses were used to determine whether TF-bound cEEs were distributed preferentially within specific evolutionary strata.

### EEs in protein structures

Amino acid sequences of cEEs in their corresponding host genes were retrieved for four species with geno2proteo[67] R package (v.0.0.6). For each gene, the cEE amino acid sequences were aligned to full-length protein sequences contained in AlphaFold[36] CIF files and MobiDB[27] entries, prioritising the UniProtKB/Swiss-Prot identifier, then UniProtKB/TrEMBL, and finally the gene name. After mapping, we computed the average AlphaFold pLDDT score (reflecting predicted structural confidence) and the MobiDB-predicted mean disorder rate (reflecting structural flexibility) for each cEE. Statistical significance of differences in these measures between the first and last deciles of cEEs was assessed using the Kruskal-Wallis test ($p < 0.05$).

### Reporting summary

Further information on research design is available in the Nature Portfolio Reporting Summary linked to this article.

## Data availability

Data supporting the findings of this study are available in a Zenodo repository (https://doi.org/10.5281/zenodo.17208730) organised by analysis block in accordance with the structure of the paper. All datasets used in this study are publicly available or have been deposited in appropriate repositories. Annotated coding exons and transcripts (human, mouse, fly, thale cress) were retrieved from UCSC and Ensembl, while FANTOM5, modENCODE, and Arabidopsis TSS data are detailed in the Methods. ChIP-seq data for exonic enhancers (ReMap2022), DNase-seq and ATAC-seq data (ENCODE, ChIP-Atlas, PlantRegMap), and STARR-seq catalogue (Supplementary Data 2 STARR-seq catalogues) were integrated to identify cEEs. G-quadruplex sequencing data (GSE110582) and the newly generated EE STARR-seq dataset (GEO accession GSE292804) were also incorporated. Cancer mutation data were obtained from the TCGA PanCanAtlas. Genomic interactions (promoter capture Hi-C), eQTLs (GTExv8), and ENCODE-rE2G mappings were used to define EE-gene relationships, while phyloP conservation scores (UCSC, PlantRegMap) and gene age classifications (Trigos etal.) further contextualised EE evolution. A genome Browser track hub containing all exonic enhancers identified in this study is available in the UCSC public hubs (https://genome.ucsc.edu/cgi-bin/hgHubConnect) and also public sessions

(https://genome.ucsc.edu/cgi-bin/hgPublicSessions). Source Data are provided with this paper. Generated plasmids, reporter constructs, and CRISPRi guide sequences generated in this study are available from the corresponding author. Source data are provided with this paper.

## Code availability

We deposited codes and bioinformatics environments in GitHub at (https://github.com/benoitballester/ExonEnhancer) and in Zenodo (https://doi.org/10.5281/zenodo.18255062). Both data and codes are publicly available for the replication of the whole study.

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

## Acknowledgements

The authors thank Robin Steinhaus for his assistance with fabian-tools and lifting the VCF query capacity. We also appreciate Science AAAS for granting permission to reproduce and modify the PanCanAtlas schema in Fig. 5c. This work was supported by a PhD Fellowship awarded to J.-C.M. from the French Ministry of Higher Education and Research (MESR), Institut National de la Santé et de la Recherche Médicale (INSERM), the Core Cluster of the Institut Français de Bioinformatique (IFB; ANR-11-INBS-0013), and by the Agence Nationale pour la Recherche (ANR; grant ANR-23-CE12-0008-01). A Marie Sklodowska-Curie Action postdoctoral fellowship (Eprom-101065610) supported A.V.O. We acknowledge the contribution of AniRA lentivectors production facility from the CELPHEDIA Infrastructure and SFR Biosciences (UAR3444/CNRS, US8/Inserm, ENS de Lyon, UCBL), especially Gisèle Froment, Aurélie Thibaut and Caroline Costa. We thank the Marseille-Luminy cell biology platform for managing cell culture and Nori Sadouni from HL BIOPROCESS (Marseille, France) for the STARR-seq preprocessing. The results presented here are based on data generated by the TCGA Research Network, the GTEx project, the ENCODE Consortium and its production laboratories, as well as independent laboratories that submitted raw ChIP-seq and other omics datasets to public repositories (GEO). We thank Andreas Zanzoni for a helpful discussion about protein disorder.

## Author contributions

B.B. conceived and supervised the project. J-C.M. developed computational methods, curated ATAC-seq, DNase I, and STARR-seq datasets, and performed data analysis. M.T. and F.G. carried out luciferase reporter assays. I.M. and M.T. conducted CRISPRi experiments. A.V.O. performed STARR-seq assays and selected and designed CRISPRi guides. S.S. supervised STARR-seq and CRISPRi experiments. J-C.M. and B.B. prepared the figures, and J-C.M., S.S., and B.B. wrote the manuscript with input from all authors.

## Competing interests

All authors declare no competing interests.
