## [Transparent Peer Review file · Nature Communications]

Exonic enhancers are a widespread class of dual-function regulatory elements

Corresponding Author: Dr Benoit Ballester

Version 0:

Reviewer comments:

Reviewer #1

(Remarks to the Author)

The manuscript by Mounren and colleagues titled “Exonic enhancers are a widespread class of dual-function regulatory elements” describes a large-scale, integrative analysis of genome regulatory data overlapping known coding regions. After defining the exonic enhancers that they will consider, the authors show that these regions have multiple signatures of enhancer activity both by comparing the regions to non-exonic enhancers and through experimental validation.

Overall, the manuscript is a valuable contribution towards the understanding of genome regulation based on its effective reuse of uniformly processed data. However, the manuscript overstates the novelty of its insights and seems to underplay previous relevant work. For example, exonic enhancers have been known to exist for nearly 30 years and have been subject to many studies including a 2011 study that analyzed a set of 29 genomes to find protein coding sequences that are under selection for overlapping functions including enhancers (Lin et al 2011). Notably, a 2016 editorial in *Genome Medicine* warning about the need to consider these when doing genome interpretation is titled “Exonic enhancers: proceed with caution in exome and genome sequencing studies” (Ahituv 2016) strongly suggesting that the “fundamental change in interpreting genetics variation in coding regions” may be well underway or is at least known the community. Neither of these publications are cited, but numerous others are cited making it clear that the authors have awareness of the existing literature. Properly placing this work in context and being clear about what aspects are novel should be a critical focus of any revision as this manuscript is simply cannot be about the discovery of exonic enhancers since they are already well known, so it must be about their genome-wide characterization.

In summary, the results reported here are useful because of the scope of the analysis undertaken and the interaction of the large uniformly processed functional dataset and validation experiments. The authors must place the work in context and be clear about what are truly novel findings, what findings extend more generally results that have been observed or suggested in a small number of loci, and what results confirm previous observations with new technologies. There is a lot in this manuscript and for those things that are not novel, they should be removed to focus on something more straightforward and more impactful.

Specific comments:

1. Naming conventions. The authors identify 13,481 human exonic regions satisfying their stringent filters for multiple transcription factor binding and designate these as “exonic enhancers”. Terminology more supported by the evidence presented is “putative exonic enhancers” or “candidate exonic enhancers”. This review will use the former term (pEE) throughout.
2. Expectation value for reported results given genome composition. Throughout the manuscript, results are reported with statements of expectation or significance. For example, on Page 2 Line 31, it is reported that 3.9% of DNase I hypersensitive sites overlap exonic regions. Is this a significant enrichment or is this what would be expected given a sequence-corrected distribution of DHS sites across the genome? Similar questions about expected overlap or significance occur in other contexts including TF binding in exons (4.2%) [Page 3 Line 4], overlap with open chromatin marks [Page 3 Line 33], binding motif occupancy [Page 4 Line 13-14], human-mouse conservation [Page 8 Line 17].

Fundamentally this is a question of whether pEEs are something common, rare or just what would happen if you have lots of

enhancers, a big genome, incomplete ascertainment of functional activities, and the need to put those enhancers somewhere. The results from various species should give insight on this question.

3. What is the expression profile of the exons with pEEs? Are these profiles different than exons without? Are they more or less likely to be alternatively spliced or part of the principal isoform? Analysis of GTEx data could be used here as well as data from the APPRIS resource. Regardless of the results of this analysis, it should give insight into the potential impact of exons subject to different evolutionary pressures function as coding regions. Note that the suggestion that the pEEs are under different evolutionary pressures is worthwhile to pursue and this analysis could give insight to that idea as well.

4. The authors state that cell- or tissue-specific considerations of the pEEs are outside the scope of this manuscript (although they do seem to have some data that could at least partly address this question from the STARR-seq data from K562 as compared to the pEEs defined in other cell types). Could the analysis of the GTEx data make this possible assuming that repeating the entire STARR-seq experiment in another cell type may not be feasible?

5. The claim that pEEs are “a distinct and functional class of enhancers” is confusing. What is being claimed to be distinct about this group? Is it only that the overlap exons or is there something else?

6. Page 4 Line 25: “many exons act as bona fide enhancers”. It seems that what has been shown in this results section is that “many exons have characteristics of bona fide enhancers”.

7. Page 5 Lines 7-12: It is not clear how to interpret the results from overlapping the pEE with gnomAD. For example, do the variants that overlap the pEE have a lower variant allele frequency than for other exons? Are gnomAD variants less likely to occur in synonymous sites in the pEE compared to non-pEE coding regions? Either of these results would suggest the pEE sites are constrained in the human population. This seems like a missed opportunity to characterize the pEEs although it may be the case that there are too few variants even in gnomAD for this analysis to be statistically significant.

8. Page 5 Line 21-22: It is not clear why it would be necessary to filter the STARR-seq assays to exclude any intron sequence. Enhancer boundaries from TF binding data are generally not at basepair resolution. A more complete justification of this should be presented.

9. Page 6 Line 16-17: Is the result that variants in strong pEEs tend to decrease in enhancer activities while those in weak pEEs simply an effect of regression to the mean? In other words, given the limitations of only synonymous variants, there are limited possible changes to the binding sites.

10. The sections on the genetic variation in the pEEs in cancer are interesting and, although may overstate the clinical relevance, are an effective way to characterize these regions.

11. Page 8 Line 19: “70% appear human-specific” is not justified given the data here. All that the data support is that 70% are not shared with mouse, which could be the effect of more limited data in mouse, which is acknowledged later in the sentence! Similarly, the shadow pEEs results are likely overstated as unmatched TF data sets define the pEE sets in each species.

12. Page 8 Line 38: The comparison between human-mouse overlap defined by hundreds/thousands TF binding experiments and human-rat / human-dog defined by four TF binding experiments is not obviously valid and does not provide “partial correlation with phylogenetic distance”. This sentence is not at all clear.

13. Page 9 Line 37: This is seemingly the first reference to “common and rare” variants and is a missed opportunity. Many of the authors claims would be strengthened or more believable if there were differences in their observations based on the allele frequency of the variants overlapping the pEEs. For example, if variants in pEEs are rarer than in exons generally, this suggests stronger constraint in the pEEs. This analysis could be done with gnomAD as well.

Other comments:

All figures should be checked and updated to ensure that they are accessible by color blind users. There are multiple places where red and green would not be distinguishable including at a minimum Figures 1g, 2b, 3d, 4a, 7b, and S10. Additionally, the placement of red next to orange in several figures can be difficult for people with normal color vision to separate.

Reviewer #3

(Remarks to the Author)

The authors present a comprehensive investigation of the phenomenon of exonic enhancers, in which specific protein coding sequences also exhibit the features of, and apparently carry out the functions of, transcriptional enhancers. They leverage multiple online databases (ReMap, etc.) to determine the proportions of coding exons (in humans and in 3 other species) that exhibit enhancer signatures (primarily TF binding, but also DNaseI hypersensitivity, ATAC-seq peaks, etc.), in the process constructing a list of ~13.4k exons (~8% of the total, in human) that represent the best enhancer candidates. These candidates overlap substantially with STARR-seq activity (both derived from existing databases and performed by the authors) and G-quadruplex sequence motifs. Tests of 20 of these EEs in transient assays show significant (>2-fold) activity for 8 of them, while transient assays of EE sequences harboring variants derived from the gnomAD database show that the variants lose enhancer activity, while introduction of silent mutations designed to increase TF binding result in

greater activity. Systematic introduction of coding variants into a STARR-seq EE library results in changes in activity, and inspection of promoter-enhancer Hi-C databases indicates that they can be linked to target promoters nearly as often as conventional enhancers. The authors intersect the EE-target promoter links they find with the TCGA Pan Cancer Atlas and determine that point mutations in defined EEs can be associated with specific tumors, insofar as they are also associated with transcriptional changes in their (putative) target genes and changes in TF binding to the enhancer regions; importantly, even synonymous mutations exhibit such correlated changes. CRISPRi of selected EEs results in decreases in expression of host and/or neighboring genes. Finally, the authors investigate the evolutionary conservation of exonic enhancers. Exonic enhancers are not a newly appreciated or otherwise novel phenomenon. They have been recognized for nearly 15 years, and at least one prior review has been dedicated to them. That review, however, is more than a decade old, and even though it (presumably) collected the sum of what was known about EEs at the time, it wasn't nearly as comprehensive or informative as this study. I'm not sure that there is a major question that might be asked regarding EEs that the authors haven't at least begun to address in this manuscript.

Comments:

--Why should the peak of any given enhancer signature occur at the midpoint of an exon (Fig. 1E)? This would seem to have implications for how such sequences evolved.

--Interesting that enhancer signatures peak just upstream of first coding exon (Figure S4), which would be expected, given the similarities between enhancers and promoters.

--In Fig. 3C, EE19 appears to have more enhancer activity on the SV40 promoter than on its actual promoter (this is not a problem for the other EE tested).

--"MobiDB27 analysis further supported this notion, showing that highly TF-bound EEs are frequently located in intrinsically disordered protein segments" – What on earth does that mean?

Reviewer #4

(Remarks to the Author)

The manuscript by Mouren and colleagues focuses on exonic enhancers. Based on re-analysis of publicly available data, the authors show that many exons have an enhancer-like signature, incl. the presence of open chromatin, active histone modifications, and transcription factor binding. Further analysis of public data regarding human genetic variation and disease predisposition as well as cancer genomes identifies correlations between variants/mutations in exonic enhancers and disease predisposition/severity. Using reporter assays, incl. a high-throughput STARR-seq assay, the authors show that a subset of the exons with an enhancer signature can drive gene expression. Furthermore, the authors show that CRISPRi perturbations of exonic enhancers reduce gene expression. Finally, the authors perform evolutionary comparisons of exonic enhancer sequences.

Together, the authors provide a comprehensive overview of enhancer signatures in exons across species and their potential association with human disease, although the functional validation of these elements could have been more thorough. The manuscript is well written and the figures are clear and easy to follow. As discussed below, the main message of the manuscript is in my view not surprising per se, but I think the analyses presented are in principle of interest for the readership of Nature Communications.

Major comments

1. Although it is obvious that exonic DNA sequences have less flexibility to evolve into a potent enhancer, there is in principle no reason to expect that exonic DNA sequences could not function as an enhancer, especially since intronic enhancers are extremely widespread and exonic DNA and intronic DNA are not fundamentally different. For this reason, the findings of the manuscript are in my view not very surprising. I find the systematic assessment interesting nonetheless and am not at all opposed to the publication of this work, but I think the authors could consider framing their work more clearly in light of these comments.

2. In line with the previous point, one would expect that exonic enhancers would overall be less potent compared to enhancers in the non-coding genome as they have less flexibility to evolve into an optimal enhancer sequence. This also seems evident from the STARR-seq comparisons (as the proportion of "Strong" exonic enhancers appears very small (5%) but the authors do not extensively discuss this).

3. Most of the analyses presented in the paper are correlative. The validations in the classic reporter assays and STARR-seq assay are nice, but it is well known that there is a poor correlation between reporter activity and in vivo activity. The best validation would be to assess the effects of synonymous point mutations in transcription factor binding sites in a native context (but I appreciated that this is a lot of work). The authors attempt to look at this in the context of cancer genomes, but given the large amount of mutations in this context, it is extremely difficult to assess the cause-consequence relationship. The CRISPRi experiments are in my view the only really informative functional assay in the manuscript. However, in these experiments, it is difficult to distinguish the effect of silencing the exonic enhancers from the effect that the recruited dCas9/KRAB complex may have on the transcribing polymerase. In this respect, it would be important to perform control experiments with gRNAs targeting exons that are not predicted to function as an enhancer. In addition, in these experiments it is not clear whether the authors are measuring nascent transcripts (with intronic primers) or stable transcripts (both are

interesting from my point of view).

Minor comments

1. Figure 1g: It is not clear what the fraction of exons and genes refers to here. What is the “total” to which these percentages refer?

2. What is the resolution of the promoter capture HI-C data? It seems unlikely that it is sufficient to distinguish interactions with the exonic enhancers and the surrounding regions. If that is indeed the case, conclusions about specific interactions should be toned down.

Version 1:

Reviewer comments:

Reviewer #1

(Remarks to the Author)

The revised manuscript by Mouren and colleagues addresses my comments in a comprehensive and successful way. In particular, the constraint analyses using gnomAD are extremely enlightening. I also very much appreciate the written clarity throughout the manuscript including the powerfully effective new second paragraph of the discussion.

The resulting work is, without doubt, the most complete and useful study of exonic enhancers. I look forward to seeing how they build on this work in the future.

Reviewer #3

(Remarks to the Author)

In response to reviewer comments, the authors have made extensive revisions to the text. For my own questions, the authors have mostly just re-written passages for clarification, although they now provide a new analysis of TF binding density across EEs to eliminate potential bias toward placing the peak in the center of the exon. Given that I was mostly leaning toward accepting the manuscript in my initial review, these clarifications and additions are sufficient for me. The other reviewers – particularly reviewer 1 – had more extensive comments than I did. While they are the best judges of the authors’ responses to those comments, I personally found the additional analyses the authors have provided to be impressive, and where new data are not offered, the reasoning behind the responses to be adequate. In particular, all 3 reviewers noted that EEs are far from being a novel topic, and so we all reacted poorly to the absence of sufficient citations of earlier studies and reviews. The revised manuscript benefits from acknowledgement of earlier work, and now provides a more accurate perspective on which aspects of the current study represent novel advancements.

Reviewer #4

(Remarks to the Author)

I would like to thank the authors for addressing my comments and for their clear and insightful responses. I have no further concerns and think that the manuscript is ready for publication in Nature Communications.

Response to reviewers and editor

We thank all three reviewers for their thoughtful and constructive critiques.

We are encouraged that, despite the constructive criticisms, **all three reviewers** recognise the value and breadth of our work. **Reviewer #1** describes the manuscript as “a *valuable contribution*” whose findings are “*useful because of the scope of the analysis undertaken.*” **Reviewer #3** notes that no major question on exonic enhancers “*hasn’t at least begun to be addressed in this study*” and calls it more “*comprehensive and informative*” than prior studies. **Reviewer #4** likewise states that our analyses “*are of clear interest for the readership of Nature Communications.*” These positive assessments reinforce our conviction that, once revised, the paper will make a strong and timely contribution to the field.

Reviewers' comments converge on several key themes we believe have been addressed in our revision.

- First, each reviewer asks that we better situate our work within the long-standing literature on exonic enhancers and clearly highlight what is novel about our large-scale, cross-species characterization rather than framing the phenomenon itself as new.
- Second, reviewers note instances where claims or terminology appear too strong; we will therefore adopt more evidence-based wording (e.g., “candidate exonic enhancers”) and moderate statements about distinctiveness or clinical impact.
- Finally, multiple reviewers request deeper or clearer statistical analyses such as enrichment relative to genomic expectation, expression and splicing patterns (GTEX/APPRIS), population-genetic constraint (gnomAD), and careful consideration of Hi-C resolution limits to solidify our conclusions.

Our revised manuscript and point-by-point responses will comprehensively address these shared concerns while incorporating additional analyses and figure improvements (**9 new** Supplementary Figures, **4 new** Supplementary Tables ; including color-blind-friendly palettes) wherever feasible.

We sincerely thank the reviewers and the handling editor for their careful reading, constructive feedback, and investment of time, all of which have significantly strengthened the clarity, robustness, and impact of our study.

Reviewer #1 (Remarks to the Author):	2
Specific comments 1.....	2
Specific comments 2.....	3
Specific comments 3.....	4
Specific comments 4.....	5
Specific comments 5.....	6
Specific comments 6.....	7
Specific comments 7.....	7
Specific comments 8.....	9
Specific comments 9.....	10
Specific comments 10.....	10
Specific comments 11.....	10
Specific comments 12.....	11
Specific comments 13.....	11
Other comments.....	13
Reviewer #3 (Remarks to the Author):	14
Comment 1.....	14
Comment 2.....	14
Comment 3.....	15
Comment 4.....	15
Reviewer #4 (Remarks to the Author):	16
Major comment 1.....	16
Major comment 2.....	17
Major comment 3.....	18
Minor comments.....	20

Reviewer #1 (Remarks to the Author):

The manuscript by Mouren and colleagues titled “Exonic enhancers are a widespread class of dual-function regulatory elements” describes a large-scale, integrative analysis of genome regulatory data overlapping known coding regions. After defining the exonic enhancers that they will consider, the authors show that these regions have multiple signatures of enhancer activity both by comparing the regions to non-exonic enhancers and through experimental validation.

Overall, the manuscript is a valuable contribution towards the understanding of genome regulation based on its effective reuse of uniformly processed data. However, the manuscript overstates the novelty of its insights and seems to underplay previous relevant work. For example, exonic enhancers have been known to exist for nearly 30 years and have been subject to many studies including a 2011 study that analyzed a set of 29 genomes to find protein coding sequences that are under selection for overlapping functions including enhancers (Lin et al 2011). Notably, a 2016 editorial in *Genome Medicine* warning about the need to consider these when doing genome interpretation is titled “Exonic enhancers: proceed with caution in exome and genome sequencing studies” (Ahituv 2016) strongly suggesting that the “fundamental change in interpreting genetics variation in coding regions” may be well underway or is at least known the community. Neither of these publications are cited, but numerous others are cited making it clear that the authors have awareness of the existing literature. Properly placing this work in context and being clear about what aspects are novel should be a critical focus of any revision as this manuscript is simply cannot be about the discovery of exonic enhancers since they are already well known, so it must be about their genome-wide characterization.

In summary, the results reported here are useful because of the scope of the analysis undertaken and the interaction of the large uniformly processed functional dataset and validation experiments. The authors must place the work in context and be clear about what are truly novel findings, what findings extend more generally results that have been observed or suggested in a small number of loci, and what results confirm previous observations with new technologies. There is a lot in this manuscript and for those things that are not novel, they should be removed to focus on something more straightforward and more impactful.

We agree that it is important to clearly distinguish novelty from confirmation. In this revision, we substantially expanded the first three paragraphs of the Introduction to cite all major single-locus and early genome-wide reports of exonic enhancer activity, thereby placing our work in context (e.g., Neznanov 1997; Lampe & Tümpel 2008; Lin 2011; Birnbaum 2012,2014; Ritter 2012; Stergachis 2013; Ahituv 2016, Chen 2023). We then clearly articulate what is new: (i) a systematic census of exonic enhancers across four phyla, (ii) exon-bounded enhancer-reporter assays confirming their activity independently of intronic sequence, (iii) integration of human population variation, cancer genomics, and clinical outcome data, and (iv) evolutionary analyses linking enhancer load to protein structural context. Where our analyses confirm previous observations, we explicitly acknowledge this (e.g., exonic TF occupancy, disease relevance) and emphasize how our uniformly processed datasets and technologies extend those findings to genome scale. We believe the revised Introduction and Discussion now make the novelty, extension, and confirmation aspects of our study transparent.

We also note that our response to Reviewer #4, major comment 1, further addresses related aspects of what makes candidate exonic enhancers noteworthy, and therefore may be of interest to Reviewer #1. For convenience, we summarize the key points below:

While prior reports have described individual exonic enhancers, (*We acknowledge that our previous Introduction required a more comprehensive presentation of prior work*), the regulatory genomics field has yet to fully integrate the idea that coding DNA can systematically encode cis-regulatory activity. In our view, this hesitation may reflect the lack of robust, genome-wide EE datasets and the absence of standardized, reusable annotations. By releasing a comprehensive and accessible catalogue, we aim to enable independent validation, reanalysis, and hypothesis generation. Whether future studies confirm, refine, or challenge our findings, we hope that this resource helps catalyze broader exploration of dual-function coding elements.

Specific comments 1

Naming conventions. The authors identify 13,481 human exonic regions satisfying their stringent filters for multiple transcription factor binding and designate these as “exonic enhancers”. Terminology more supported by the evidence presented is “putative exonic enhancers” or “candidate exonic enhancers”. This review will use the former term (pEE) throughout.

We acknowledge the reviewer's preference regarding terminology.

In the revised manuscript, we now refer to our computationally defined set as "**candidate exonic enhancers (cEEs)**" to reflect that most loci await experimental validation. This designation is used consistently throughout the main text, figure legends, axis labels, and supplementary materials (figures and legends), with the acronym cEEs introduced at first mention.

The unqualified term "exonic enhancers (EEs)" is retained solely in three contexts: (i) when discussing the general concept (e.g., in the Abstract and Introduction), (ii) when referring to elements experimentally validated in this study (via luciferase or STARR-seq assays), and (iii) when citing previously published, functionally characterized examples.

We have opted not to modify the manuscript title, which is intended to reflect the broader conceptual framework rather than solely the computational set.

Specific comments 2

Expectation value for reported results given genome composition. Throughout the manuscript, results are reported with statements of expectation or significance. For example, on Page 2 Line 31, it is reported that 3.9% of DNase I hypersensitive sites overlap exonic regions. Is this a significant enrichment or is this what would be expected given a sequence-corrected distribution of DHS sites across the genome? Similar questions about expected overlap or significance occur in other contexts including TF binding in exons (4.2%) [Page 3 Line 4], overlap with open chromatin marks [Page 3 Line 33], binding motif occupancy [Page 4 Line 13-14], human-mouse conservation [Page 8 Line 17].

Fundamentally this is a question of whether pEEs are something common, rare or just what would happen if you have lots of enhancers, a big genome, incomplete ascertainment of functional activities, and the need to put those enhancers somewhere. The results from various species should give insight on this question.

We appreciate the reviewer's call for a rigorous evaluation of whether the observed overlaps exceed genomic expectations.

We now provide Supplementary Table S1, which summarises permutation-based enrichments for every feature (DHS, TF-ChIP peaks, open-chromatin marks, motif hits, and human-mouse cEE conservation) in all four species (10 000 GC-, length- and chromosome-matched shuffles; see "Permutation-based enrichment tests" in Methods).

In human, for example, 3.9 % of DNase I hypersensitive sites intersect coding exons versus 3.08 % expected ($1.26 \times$, $P = 0$), and 0.50 % of cEEs overlap DHS versus 0.39 % expected ($1.30 \times$, $P = 0$). Also, Human-mouse overlap: 28 % of human cEEs have orthologous mouse cEEs, versus 6.94 % expected from random exon sets; 4.03-fold enrichment, $P=0$. All permutation-based enrichments are summarized in Supplementary Table S1.

Equivalent tables for mouse, fly and *Arabidopsis* are included. As anticipated for compact genomes (*D. melanogaster*, *A. thaliana*), where elements densely cover the genome, the dynamic range of the permutation test is reduced. Nevertheless, we applied the same framework and reported the results, together with a cautionary note on interpretability, in Supplementary Table S1.

To enable direct comparison with earlier work (e.g., Stergachis et al., Science 2013), Figure 1b retains the simple exon-overlap plot; our values are highly concordant with those published (Stergachis et al., Science 2013), while Table S1 provides the sequence-corrected expectations requested by the reviewer.

Finally, the reviewer point is specifically useful in asking whether candidate exonic enhancers (cEEs) show unusual overlap patterns; we performed a parallel analysis restricted to cEEs (instead of all coding exons). Those results are also included in Supplementary Table S1; they confirm that cEEs exhibit the strongest enrichments of all exon subsets.

These additions provide the requested sequence-corrected expectations, demonstrate that our enrichment claims are statistically robust, and not an artefact of genome composition.

Below is a subset of Supplementary Table S1 for human features, the full table is in the Supplementary document. Main manuscript Page 4 Line 120.

Feature (human)	Observed (%) (with coding exons or cEE when specified)	Expected (%) (matched permutations)	Fold-enrichment	Empirical P
DHS	3.90	3.0841	1.26×	0
TF-ChIP peaks	4.20	2.7441	1.53×	0
H3K27ac (GTEx)	0.6391	0.6100	1.05×	2×10^{-4}
H3K27ac (ENCODE)	0.9026	0.8770	1.03x	0
H3K4me1 (ENCODE)	0.8847	0.8090	1.09×	0
H3K4me2 (ENCODE)	8.0721	1.0881	7.42x	0
Motif occupancy (cEEs)	15.640	14.751	1.06×	0
Human–mouse cEEs overlap	28.0	6.94	4.03×	0
DHS (cEEs only)	0.5033	0.3881	1.30×	0
TF peaks (cEEs only)	0.5420	0.4593	1.18×	0

Specific comments 3

What is the expression profile of the exons with pEEs? Are these profiles different than exons without? Are they more or less likely to be alternatively spliced or part of the principal isoform? Analysis of GTEx data could be used here as well as data from the APPRIS resource. Regardless of the results of this analysis, it should give insight into the potential impact of exons subject to different evolutionary pressures function as coding regions. Note that the suggestion that the pEEs are under different evolutionary pressures is worthwhile to pursue and this analysis could give insight to that idea as well.

We have extended the manuscript with three complementary analyses to assess the expression profiles of exons with candidate exonic enhancers (cEEs) and exons without (controls), as well as, their splicing patterns. The question behind being: are they any different ?

The analyses are as below :

(i) Transcript-isoform prevalence (existing Supplementary Fig. S8, and updated). Here we point the reviewer to existing Fig. S8 (referenced in the main text page 3) in which we show that across all four species, candidate exonic enhancers (cEEs) are incorporated into most transcript isoforms of their host genes.

This Figure S8 shows that cEE are present in 92.65 % (mean across the four species) of a gene's transcripts similar to 90,57% for matched control exons (TOST $P < 10e-16$; here for *D.mel*, which represents the least significant comparison; all others are lower). This indicates that cEEs are rarely confined to minor isoforms, and are, as controls exons, as prevalent in constitutive transcripts.

(ii) Constitutive vs alternative splicing (new Supplementary Fig. S9). Using GTEx v8 exon PSI values (PSI values taken from Einson et al. 2023, we observed that 88.3 % of cEEs (across all tissues) are constitutively included (PSI ≥ 0.9 in 100% of tissues) similar to 83.52 % of controls (TOST $P=0$). This analysis is limited to 18 tissues, which were chosen for their coverage in GTEx and their coverage of the most coding genes possible.

(iii) Principal-isoform enrichment (new Supplementary. Fig. S10). Cross-referencing APPRIS v50 reveals that:

- H.sap: 71.89 % of cEEs reside in principal isoforms (PM/P1/P2) versus 73.2 % of controls ($P=1.3e-10$).
- M.mus: 69.88 % of cEEs reside in principal isoforms (PM/P1/P2) versus 74.06 % of controls ($P=9.7e-31$).
- D.mel: 75.3 % of cEEs reside in principal isoforms (PM/P1/P2) versus 81.05 % of controls ($P=8.2e-13$).

Although APPRIS annotation distributions between cEEs and Ctrl- differ statistically (Chi-squared test, $p < 0.001$), the effect sizes are small (Cramér's $V = 0.05-0.07$), suggesting similar overall annotation profiles between groups. In other words, the APPRIS comparison shows that the proportion of cEEs falling in principal isoforms (PM/P1/P2) is only a few percentage points lower than in control exons; despite highly significant $\chi^2 P$ values, the Cramér's V values indicate very small effect sizes.

Collectively, the new analyses (together with existing Fig. S8) demonstrate that cEEs are preferentially located in highly expressed, constitutive exons of principal isoforms, supporting the idea that they are similar to ordinary coding exons. Main manuscript Page 3 Lines 107-110.

We agree with the reviewer that looking at the expression of exons with candidate exonic enhancers (cEEs) and splicing context would be worthwhile to pursue for the understanding of the different selective pressures on those cEEs. However we believe that this analysis belongs to a larger question which we plan to pursue in another project.

Specific comments 4

The authors state that cell- or tissue-specific considerations of the pEEs are outside the scope of this manuscript (although they do seem to have some data that could at least partly address this question from the STARR-seq data from K562 as compared to the pEEs defined in other cell types). Could the analysis of the GTEx data make this possible assuming that repeating the entire STARR-seq experiment in another cell type may not be feasible?

Evidence from our K-562 STARR-seq screen

Supplementary Figure S17 already hints at cell-type restriction. Our K-562 library included 479 candidate exonic enhancers (cEEs) that are bound by transcription factors assayed in two non-haematopoietic cell lines (A549 and GM12878). These "A549" and "GM12878" cEEs drive markedly weaker reporter activity in K-562 than cEEs whose ChIP signatures originate in K-562 itself (Student t-test, $P < 1 \times 10^{-5}$). The result indicates that at least a subset of cEEs behaves in a cell-line-restricted manner. A full survey will require re-running STARR-seq in additional cell types; a grant proposal to perform that experiment is currently under review.

Clarification : GTEx can only be a proxy

Regarding the referee's proposed GTEx analysis, please note that this analysis necessarily reflects the gene's host-exon expression patterns (or targets), not direct enhancer output. Thus any tissue signal we derive from GTEx reflects the context in which the gene is transcribed, not direct activity of the cEE. Dissecting enhancer activity itself will ultimately need functional assays such as STARR-seq or CRISPRi in multiple tissues, a scale of work that lies beyond the present revision.

GTEx tissue expression of cEE host genes (Supplementary Fig. S18-S19)

Despite this limitation, we carried out a GTEx-based analysis as the referee requested (Methods subsection "GTEx tissue-specific expression of cEE-containing exons"). In brief, we assigned each cEE a "biotype signature" from its dominant ChIP-seq tissue, then examined host-gene (or target-gene) TPMs across 18 GTEx tissues. Violin plots show that some signatures (e.g. Reproductive_M) are more expressed in their matching tissue, whereas others (e.g. Respiratory) are broadly expressed. The accompanying heatmaps for both host and predicted target genes reinforce this mixed picture: some tissue-matched enrichment can be visible, but most cEEs sit in genes active in several organs. Overall, the behaviour seems to parallel that of canonical enhancers, some of which are ubiquitous while others are sharply tissue-restricted. However, as explained above, GTEx data

can only report on the expression breadth of host or target genes, not on direct enhancer activity. We therefore refrain from concluding on tissue specificity of cEEs themselves from this analysis, and interpret the GTEx results only as a proxy context for the genes in which cEEs reside.

Uhlén/HPA tissue-specificity classification (Supplementary Fig. S20 panel a-d)

To place the GTEx observed trends in a second framework we used the Human Protein Atlas categories of Uhlén et al. Science (2019) (Supplementary Fig. S20 panel a, b). Uhlén et al. integrated GTEx and Human Protein Atlas RNA-seq data to assign every protein-coding gene to one of several tissue-specificity classes (e.g., “tissue enriched,” “group enriched,” “tissue enhanced,” “low specificity,” “not detected”). We then employed those classes into “Elevated expression specificity” or “Low tissue specificity” groups, as described in Uhlén et al. Science (2019). Genes associated with cEEs frequently exhibit expression patterns enriched in the tissue corresponding to the cEE’s biotype signature.

Using the Uhlén/HPA scheme collapsed into two classes (“elevated expression in ≥ 1 tissue” vs “low tissue specificity”), cEE-associated genes (both hosts and predicted targets) show a slight enrichment for the “elevated expression” class relative to matched control exons and the genome-wide background (Supplementary Fig. S20 panel c; $\chi^2 = 2 \times 10^{-66}$, Cramér’s $V = 0.067$). The effect is statistically robust but modest in magnitude, indicating that cEEs are linked to genes spanning a broad specificity spectrum: many display elevated expression in at least one tissue, yet a substantial fraction remain broadly expressed.

Likewise, the Yanai τ index calculated from Uhlén et al. (2016) shows a slightly lower median for cEE hosts (0.43, Supplementary Fig. S20 panel d) than for matched controls (0.45), supporting the view that, on average, cEEs reside in genes expressed across a broader tissue panel, although strongly tissue-specific examples do exist.

ENCODE cell-line RNA-seq for STARR-validated cEEs (Supplementary Fig. S20 panel e-f)

Finally, we asked whether the “strong” K-562 cEEs identified by STARR-seq live in genes preferentially expressed in K-562 relative to A549 and GM12878. For these K-562 cEEs, we retrieved the expression levels (FPKM) of their host genes and, where available, their predicted target genes in each oA-549, and GM12878 cell lines. Host- and target-gene FPKMs are indeed highest in K-562, median expression was highest in K-562, with pairwise comparisons showing only a trend toward significance (lowest $P = 1.308 \times 10^{-2}$ for K-562 vs A-549; $P = 0.206$ for K-562 vs GM12878). The trend aligns with the STARR-seq result and again suggests cell-line bias for a subset of cEEs.

Conclusion

Taken together, STARR-seq, GTEx, Uhlén/HPA and ENCODE RNA-seq all point to a mixed landscape: many cEEs gene host are used broadly, whereas others are clearly tissue- or cell-type-restricted, mirroring the spectrum described for classical enhancers. These analyses address the reviewer’s request without new wet-lab experiments, while motivating a dedicated, multi-cell-line functional study in future work.

We hope these extended analyses address the referee’s point.

We have added the corresponding text on Page 6 Line 226

“Used only as a host-gene proxy (not direct enhancer output), complementary GTEx/ENCODE transcriptomes analyses indicate a mixed landscape: cEEs associate with both broadly and narrowly expressed genes, paralleling canonical enhancer behaviour (Supplementary Figs. 18–20).”

The corresponding methods sections “**GTEx, Uhlén-based, ENCODE RNA-seq analyses**” have been added to the manuscript.

Specific comments 5

The claim that pEEs are “a distinct and functional class of enhancers” is confusing. What is being claimed to be distinct about this group? Is it only that the overlap exons or is there something else?

Our intention was not to suggest that candidate exonic enhancers (cEEs) constitute a mechanistically separate enhancer type, but rather that they form a recognisable subclass of enhancers distinguished by their genomic context (protein-coding exons) and by, possibly, a characteristic combination of molecular and evolutionary features.

To avoid ambiguity, we have replaced the phrase “a distinct and functional class of enhancers” with “a **subclass of enhancers embedded within coding exons**” in the text (Page 4 Line 147).

Specific comments 6

Page 4 Line 25: “many exons act as bona fide enhancers”. It seems that what has been shown in this results section is that “many exons have characteristics of bona fide enhancers”.

We agree that the wording was too assertive at this point in the Results, where the evidence presented is correlative (TF binding, chromatin marks, motif density) rather than functional. We have therefore revised the sentence to reflect the strength of the data (using “display”, rather than “act”):

Revised manuscript line (Page 5, now Line 155):

*“In conclusion, our multi-species analyses confirm that many exons **display** multiple hallmark characteristics of enhancer, characterized by open chromatin, STARR-seq activity, relevant TF binding sites, and an enrichment of G4 and GC-rich sequences.”*

This change ensures the language used in the manuscript is proportional to the data shown at each stage and fully addresses the reviewer’s concern.

Specific comments 7

Page 5 Lines 7-12: It is not clear how to interpret the results from overlapping the pEE with gnomAD. For example, do the variants that overlap the pEE have a lower variant allele frequency than for other exons? Are gnomAD variants less likely to occur in synonymous sites in the pEE compared to non-pEE coding regions? Either of these results would suggest the pEE sites are constrained in the human population. This seems like a missed opportunity to characterize the pEEs although it may be the case that there are too few variants even in gnomAD for this analysis to be statistically significant.

We have summarized the referee's question as : “Please clarify whether variants that overlap pEEs show signs of stronger constraint in gnomAD”. The answers below are also linked with the Specific comment 13.

Below we list the new analyses and where they appear in the manuscript:

Methods: A new subsection “*gnomAD-based constraint analysis of cEEs*” details the pipeline for all eight metrics listed below.

Results : After much attempt to include those results (for comments 7 and 13), we decided to add them into a separate new paragraph entitled *Population genetics and selection on Exonic Enhancers page 6 lines 196-213*.

Figure and Data: All numerical results are compiled in **Supplementary Table S2**; the synonymous-fraction violin plot is **Supplementary Fig. S16**; the TF-decile allele-frequency plot is **Supplementary Fig. S21**.

The metrics we have measured :

Using gnomAD v3.12 whole-genome SNVs (~76k unrelated individuals) we compared cEEs to control exons (Ctrl-) with eight complementary population-genetic and evolutionary metrics. For clarity we summarize these metrics below :

Short-term polymorphism	Long-term divergence	Composite tests
 • Median allele frequency (AF) • Fraction of ultra-rare SNVs (AF ≤ 0.01 %) • Synonymous SNVs per kb • Synonymous / total variant ratio • pN/pS (missense / synonymous) 	 • dN/dS in human–mouse orthologues 	 • Direction of Selection (DoS) • Fixation Index (FI)

All statistics were evaluated with two-sided Mann-Whitney tests (per-exon medians) or χ^2 on raw counts (pN/pS, dN/dS, AF bins).

Results added in Supplementary Table S2 :

Metric (gnomAD v.3.12)	cEEs (n = 870,139)	Ctrl- (n = 236,177)	Test
Median AF of all variants	6.6e-06	8.0e-06	Mann-Whitney $P = 0$
Ultra-rare fraction (AF $\leq 0.01\%$) (Values are percentages of the total variant count in each set)	90.9 %	90.4 %	$\chi^2 P = 1.8 \times 10^{-13}$
Median nb of synonymous SNVs normalized per exon size	0.065	0.042	Mann-Whitney $P = 0$
Median Synonymous/All variant ratio	0.329	0.265	Mann-Whitney $P = 0$
pN/pS (gnomAD missense / synonymous)	All: 0.584 per-exon median: 0.587	All: 0.611 per-exon median: 0.551	$\chi^2 P = 1.3e-137$ Mann-Whitney $P = 1.09e-21$
dN/dS (human-mouse orthologues)	All: 0.427 per-exon median: 0.086 n=7,802	All: 0.311 per-exon median: 0.088 n=8,095	$\chi^2 P = 2.1e-13$ Mann-Whitney $P = 6.2e-4$
Direction of Selection (DoS) median	0.042 n=7,298	-0.083 n=5,982	Mann-Whitney $P = 4.2e-39$
Fixation Index (FI) median	1.235 n=7,298	0.666 n=5,982	Mann-Whitney $P = 4.7e-43$

Interpretation (Supplementary Table S2; Fig. S16, S21)

Short-term constraint. cEEs display lower median allele frequencies and a larger fraction of ultra-rare SNVs than control exons. They also harbour proportionally more synonymous and fewer missense changes, reflected also in the violin plot of the synonymous/total ratio (median 0.33 vs 0.27; **Fig. S16**), and the reduced pN/pS (0.59 \rightarrow 0.62), which are classic signatures of strong purifying selection in contemporary humans.

Long-term signal. The human-mouse dN/dS is modestly higher for cEEs (0.43 \rightarrow 0.31), suggesting that over deep evolutionary time amino-acid substitutions have accumulated somewhat more freely, perhaps through compensatory changes that keep embedded regulatory motifs intact.

Composite tests. A positive Direction-of-Selection (DoS = +0.04) and FI > 1 (median 1.24) indicate a mild excess of fixed nonsynonymous substitutions relative to polymorphism, consistent with episodic adaptive or compensatory evolution layered on top of strong background constraint.

New paragraph added to Results Page 6 Lines 196-213:

“Population genetics and selection on Exonic Enhancers

To explore how coding variation shapes exonic-enhancer function, we first interrogated population and evolutionary data for cEEs. Using gnomAD28, we observed that cEEs carry rarer variants and harbour a higher synonymous fraction, yielding a lower missense-to-synonymous ratio (pN/pS = 0.59 vs 0.62; $\chi^2 P = 1.3 \times 10^{-137}$) and more synonymous changes (0.33 vs 0.27; Mann-Whitney $P < 1 \times 10^{-16}$; Supplementary Fig. 16; Supplementary Table 2). Stratifying by allele-frequency bins

shows a clear skew toward ultra-rare variants in cEE (AF $\leq 0.01\%$: 90.93% vs 90.44%; two-proportion z-test $P = 1.82 \times 10^{-13}$; χ^2 across bins $P = 2.5 \times 10^{-31}$; Supplementary Table 3). While SNP density increases with TF-binding load, the allele-frequency spectrum is essentially unchanged across TF deciles (~91% ultra-rare, 6% rare, 2% low-frequency, 1% common; Supplementary Table 3), indicating that purifying selection acts broadly across cEEs rather than being confined to the most TF-dense exons (Supplementary Fig. 21-22). cEEs also display a slightly higher human–mouse dN/dS together with positive values for the Direction-of-Selection statistic (DoS) and the Fixation Index (FI), metrics that compare divergence to polymorphism, suggesting that compensatory coding changes have accumulated over deep evolutionary time (Supplementary Table 2 and Supplementary Fig. 21-22). Overall, these results indicate that cEEs appear constrained at the population level yet evolve via compensatory substitutions over deep time. Given these constraints, we next quantified cEE activity and variant effects at scale using STARR-seq. “

We believe these analyses fully address the reviewer’s concerns and substantially strengthen the manuscript.

Specific comments 8

Page 5 Line 21-22: It is not clear why it would be necessary to filter the STARR-seq assays to exclude any intron sequence. Enhancer boundaries from TF binding data are generally not at basepair resolution. A more complete justification of this should be presented.

Our goal with the STARR-seq validation was to determine whether the exonic sequence alone can act as an enhancer.

Public STARR-seq libraries tile the genome with 200- to 500-bp fragments that frequently extend beyond exon boundaries. Intronic DNA harbours a high density of enhancers (clusters of ChIP-seq peaks) and cCREs. Consequently, a positive STARR-seq signal from a hybrid fragment that contains intronic nucleotides cannot be unambiguously attributed to the exon.

To eliminate this confounding factor we applied a conservative filter: only fragments whose coordinates were entirely contained within annotated coding-exon boundaries were retained. This strategy guarantees that any measured activity derives from exonic sequence and not from a flanking intronic enhancer that happens to fall within the amplicon. While TF-binding peaks do not provide exact enhancer borders, including intronic sequence would systematically bias the assay toward detecting intronic, rather than exonic, regulatory activity.

We believe this clarification fully explains why intronic sequence was excluded and ensures that the STARR-seq results genuinely reflect enhancer potential intrinsic to the exon.

An explanatory sentence has been added to the **Results** (Page. 6, lines 217-220):

“To assay genome-wide enhancer activity, we screened a STARR-seq library of 2,068 exon-bounded cEE fragments (no intronic sequence) in K-562 cells alongside positive and negative controls (Fig. 4a, Methods), ensuring that any signal originates from coding DNA rather than neighbouring intronic elements.

The **Methods** section “EE STARR-seq processing and analysis : Selection criteria” has been expanded to include this rationale (Page 18 Line 679):

“For each candidate exonic enhancer (cEE) we synthesised only the annotated coding-exon sequence, trimming precisely at exon boundaries and excluding all flanking intronic DNA, before cloning the fragment into the STARR-seq vector. This exon-only design prevents intronic enhancers, which can be adjacent to coding regions, from confounding interpretation of the reporter signal.

We trust that these clarifications address the reviewer’s concern.

Specific comments 9

Page 6 Line 16-17: Is the result that variants in strong pEEs tend to decrease in enhancer activities while those in weak pEEs simply an effect of regression to the mean? In other words, given the limitations of only synonymous variants, there are limited possible changes to the binding sites.

We believe that the referee is asking “*Is the decrease seen in strong cEEs (and increase in weak ones) simply regression to the mean?*”

To test this possibility, we modelled the effect expected from experimental noise alone. We implemented a simulation-based null model accounting for experimental noise. For each wild-type (WT) insert with at least three technical replicates, we calculated the mean \log_2 -transformed enhancer activity (\log_2FC_{WT}) and its replicate standard deviation (σ_i), representing the expected noise. Observed variant effects were defined as $\Delta A = \log_2FC_{variant} - \log_2FC_{WT}$, and regressed against \log_2FC_{WT} to estimate the empirical slope (β_{obs}).

To generate the null distribution, we simulated 10,000 pseudo-variants by drawing a synthetic activity value A_{sim} from a normal distribution $N(\log_2FC_{WT}, \sigma_i^2)$ for each WT insert. For each simulation, we calculated $\Delta A_{sim} = A_{sim} - \log_2FC_{WT}$ and regressed it on \log_2FC_{WT} , recording the slope (β_{sim}). The empirical P-value was defined as the proportion of simulations where $\beta_{sim} \leq \beta_{obs}$. This approach preserves the variance structure of each insert while testing the null hypothesis that observed variant effects arise solely from measurement noise.

As shown in Supplementary Fig. S23, the observed slope lay far in the negative tail, indicating that the effect cannot be explained by noise alone. This supports a genuine directional bias consistent with motif-disrupting or motif-enhancing effects of synonymous variants, rather than artefactual regression to the mean.

We added a sentence in the **Results** section “Coding variants modulate Exonic Enhancers activity” on page 7 line 258:

“Simulations confirmed that this negative correlation far exceeds what would be expected from regression to the mean (Supplementary Fig. 23).”

We added a new method section titled “**Regression-to-the-mean simulation**” Page 21 Line 792.

Specific comments 10

The sections on the genetic variation in the pEEs in cancer are interesting and, although may overstate the clinical relevance, are an effective way to characterize these regions.

We appreciate your interest in this section. Our initial writing was purposely very cautious, using words such as “susceptible”, “potentially”. However the reviewer perceived a risk of over-interpretation. We thus applied a tonal adjustment plus one clarifying sentence to defuse the concern without weakening the scientific message.

Original text (Results, last paragraph of ‘Somatic mutation landscape of cEEs’):

“Collectively, these findings indicate that EEs are susceptible to pathological rewiring in cancer, potentially influencing disease progression through the dysregulation of critical target genes.”

Revised text Page 8 Line 308:

*“Collectively, these findings indicate that **a subset of candidate exonic enhancers (cEEs) are somatically mutated in tumours and that such mutations are statistically associated with altered expression of nearby genes.** Although causal links to tumour progression remain to be demonstrated experimentally, these observations highlight cEEs as genomically and transcriptionally responsive loci in cancer.”*

We believe these revisions address the reviewer’s concern while preserving the key scientific point.

Specific comments 11

Page 8 Line 19: “70% appear human-specific” is not justified given the data here. All that the data support is that 70% are not shared with mouse, which could be the effect of more limited data in mouse, which is acknowledged

later in the sentence! Similarly, the shadow pEEs results are likely overstated as unmatched TF data sets define the pEE sets in each species.

We agree that our original wording implied stronger species specificity than the available data support. While the ReMap TF catalogues for human and mouse do share overlap (412 transcriptional regulators are common to both), the overall TF coverage and data volume differs between species. However we do have previous experience with these type of conservation analyses (Science 2010, Elife 2014).

To better reflect these limitations, we have revised the language in the manuscript. Specifically, we replaced “human-specific” with the more neutral “**not shared with mouse cEEs**,” and modified our terminology to avoid implying definitive absence or evolutionary novelty.

1. **Revision of manuscript wording**

Original (p. 9): “The remaining 70% appear human-specific...”

Revised (p.9 Line 345): “The remaining 70% appear not shared with mouse cEEs...”

Equivalent edits were also made in the associated figure and legend.

2. **Renaming and toning down the “shadow cEE” claims**

To avoid overinterpretation, we replaced the term “shadow cEEs” with “putative orthologous cEEs.” The revised text now reads (p. 9 Line 347):

*“Remarkably, **even the mouse exons aligned to these non-shared human cEEs**, still display low but detectable TF occupancy exceeding negative controls, suggesting emergent or sub-threshold enhancer potential. These **putative orthologous cEEs** could represent ancestral exonic enhancers that have partially decayed or remain undetected in some lineages [...]*”

We hope these changes address the referee’s concerns and better reflect the evolutionary and technical nuances of the data.

Specific comments 12

Page 8 Line 38: The comparison between human-mouse overlap defined by hundreds/thousands TF binding experiments and human-rat / human-dog defined by four TF binding experiments is not obviously valid and does not provide “partial correlation with phylogenetic distance”. This sentence is not at all clear.

We agree, the original sentence was unclear and could be misinterpreted. To clarify:

In Fig. 7d, we combine two sources of information. First, the red asterisk marks the proportion of cEEs shared between human and mouse as identified in this study (Fig. 7a), using the full ReMap TF catalogues (thousands of ChIP-seq experiments per species). This provides an estimate of conservation based on dense TF-binding data.

Second, the colored dots are derived from the reanalysis of a previous multi-species ChIP-seq study (Ballester et al., elife 2014), which profiled four transcription factors (CEBPA, FOXA1, HNF4A, ONECUT1) in liver tissue across five mammals (human, macaque, mouse, rat, dog). Although these comparisons involve fewer TFs, the experiment was tightly controlled across species and allows for matched-tissue, multi-species assessment of conservation patterns.

By overlaying these two datasets, we do not claim precise equivalence across comparisons. Rather, the multi-species Ballester et al. dataset provides a **complementary perspective**, showing that even with limited TFs, the proportion of shared cEEs roughly follows **phylogenetic distance from humans**, with higher conservation in macaques than in rats or dogs.

On page 9 we have revised most of the second paragraph (“Evolutionary dynamics and selective pressure on Exonic Enhancers” p. 9 Lines 355-365) to better reflect this interpretation and removed the ambiguous phrase containing “partial correlation with phylogenetic distance.”

Specific comments 13

Page 9 Line 37: This is seemingly the first reference to “common and rare” variants and is a missed opportunity. Many of the authors claims would be strengthened or more believable if there were differences in their

observations based on the allele frequency of the variants overlapping the pEEs. For example, if variants in pEEs are rarer than in exons generally, this suggests stronger constraints in the pEEs. This analysis could be done with gnomAD as well.

We thank the reviewer for this suggestion. Here is the analyses we have done: We have stratified gnomAD v3.1.2 single-nucleotide variants that overlap coding exons into four bins : ultra-rare (AF ≤ 0.01 %), rare (0.01–0.1 %), low-frequency (0.1–1 %), and common (> 1 %). The results are summarised in Supplementary Table S3 (see below).

Allele-frequency bin (gnomAD v.3.12)	cEEs (gnomAD n = 870,139)	Ctrl- (gnomAD n = 236,177)	Two-Proportion Z-Test
Ultra-rare (AF ≤ 0.01%)	90.93 %	90.44 %	P=1.82e-13
Rare (0.01% < AF ≤ 0.1%)	6.00 %	6.04 %	n.s.
Low-freq (0.1% < AF ≤ 1%)	1.87 %	2.06 %	P=9.51e-09
Common (AF > 1%)	1.18 %	1.44 %	P=2.94e-25

Although the absolute shifts are modest (~0.5 % of variants move from the low-frequency/common bins into the ultra-rare bin), the χ^2 statistic on the raw counts ($\chi^2 = 140.9$, d.f. = 2, $P = 2.5 \times 10^{-31}$) and a collapsed rare-vs-non-rare Fisher's exact test (odds ratio = 1.15, 95 % CI = 1.12–1.18, $P = 2.9 \times 10^{-28}$) both confirm a highly significant enrichment of very-low-frequency alleles within cEEs.

This skew towards ultra-rare variants is consistent with stronger purifying selection acting on cEE sequence relative to control coding exons.

Consistent with stronger regulatory involvement, germline SNP density (gnomAD; normalised by exon length) is modestly but significantly elevated in cEEs relative to negative-control exons and is comparable to classic intergenic enhancers; (Supplementary Fig. S21a; $P < 1 \times 10^{-4}$). In tumours, somatic variant density (TCGA; length-normalized) is likewise higher in cEEs than in matched negative-control exons (Supplementary Fig. S21b; $P < 1 \times 10^{-4}$). These patterns indicate that cEEs are more variant-rich than ordinary coding exons in both germline and cancer genomes, consistent with TF-dense exons being exposed to (or tolerating) greater sequence change.

In addition to the allele-frequency stratification (ultra-rare to common) across all cEEs and control exons, we further examined how variant frequency distribution changes as a function of transcription factor (TF) binding density. Specifically, we stratified cEEs into deciles by increasing number of distinct TFs bound and calculated the proportion of variants falling into each frequency bin (ultra-rare, rare, low-frequency, and common) using gnomAD data (Supplementary Fig. S22).

These plots show that **the allele-frequency spectrum is essentially constant across TF-binding load**: ultra-rare variants remain ~ 91 %, rare ~ 6 %, low-frequency ~ 2 %, and common ~ 1 % in every decile. What changes with TF occupancy is the **absolute number of variants** (n values above each bar), which rises from ~41 k in the lowest decile to ~198 k in the highest (also in Supplementary Fig. S22). Thus, highly bound cEEs harbour more segregating sites, but the same strong skew toward very-low-frequency alleles is maintained.

New paragraph added to Results Page 6 Lines 196-213:

“Population genetics and selection on Exonic Enhancers

To explore how coding variation shapes exonic-enhancer function, we first interrogated population and evolutionary data for cEEs. Using gnomAD28, we observed that cEEs carry rarer variants and harbour a higher synonymous fraction, yielding a lower missense-to-synonymous ratio (pN/pS = 0.59 vs 0.62; $\chi^2 P = 1.3 \times 10^{-137}$) and more synonymous changes (0.33 vs 0.27; Mann–Whitney $P < 1 \times 10^{-16}$; Supplementary Fig. 16; Supplementary Table 2). Stratifying by allele-frequency bins shows a clear skew toward ultra-rare variants in cEE (AF ≤ 0.01%: 90.93% vs 90.44%; two-proportion z-test $P = 1.82 \times 10^{-13}$; χ^2 across bins $P = 2.5 \times 10^{-31}$; Supplementary Table 3).

While SNP density increases with TF-binding load, the allele-frequency spectrum is essentially unchanged across TF deciles (~91% ultra-rare, 6% rare, 2% low-frequency, 1% common; Supplementary Table 3), indicating that purifying selection acts broadly across cEEs rather than being confined to the most TF-dense exons (Supplementary Fig. 21-22). cEEs also display a slightly higher human–mouse dN/dS together with positive values for the Direction-of-Selection statistic (DoS) and the Fixation Index (FI), metrics that compare divergence to polymorphism, suggesting that compensatory coding changes have accumulated over deep evolutionary time (Supplementary Table 2 and Supplementary Fig. 21-22). Overall, these results indicate that cEEs appear constrained at the population level yet evolve via compensatory substitutions over deep time. Given these constraints, we next quantified cEE activity and variant effects at scale using STARR-seq. “

Text added to Discussion Page 12 L434.

“Consistent with genome-wide constraint, cEEs carry a marked excess of ultra-rare variants, and although variant counts rise with TF load, the allele-frequency spectrum is nearly unchanged across TF deciles.”

We hope these analyses addressed the reviewer’s comment.

Other comments

All figures should be checked and updated to ensure that they are accessible by color blind users. There are multiple places where red and green would not be distinguishable including at a minimum Figures 1g, 2b, 3d, 4a, 7b, and S10. Additionally, the placement of red next to orange in several figures can be difficult for people with normal color vision to separate.

For those specifically listed figure panels, we adapted the red/green contrasts to colorblind (Deuteranopia) users, replacing the red, used throughout the paper to characterize human cEEs, to blue. We used the Sim Daltonism application, allowing us to “see” those contrasts as colorblind users. All figures were re-examined with the Sim Daltonism “Deuteranopia” filter to confirm adequate contrast.

Reviewer #3 (Remarks to the Author):

The authors present a comprehensive investigation of the phenomenon of exonic enhancers, in which specific protein coding sequences also exhibit the features of, and apparently carry out the functions of, transcriptional enhancers. They leverage multiple online databases (ReMap, etc.) to determine the proportions of coding exons (in humans and in 3 other species) that exhibit enhancer signatures (primarily TF binding, but also DNaseI hypersensitivity, ATAC-seq peaks, etc.), in the process constructing a list of ~13.4k exons (~8% of the total, in human) that represent the best enhancer candidates. These candidates overlap substantially with STARR-seq activity (both derived from existing databases and performed by the authors) and G-quadruplex sequence motifs. Tests of 20 of these EEs in transient assays show significant (>2-fold) activity for 8 of them, while transient assays of EE sequences harboring variants derived from the gnomAD database show that the variants lose enhancer activity, while introduction of silent mutations designed to increase TF binding result in greater activity. Systematic introduction of coding variants into a STARR-seq EE library results in changes in activity, and inspection of promoter-enhancer Hi-C databases indicates that they can be linked to target promoters nearly as often as conventional enhancers. The authors intersect the EE-target promoter links they find with the TCGA Pan Cancer Atlas and determine that point mutations in defined EEs can be associated with specific tumors, insofar as they are also associated with transcriptional changes in their (putative) target genes and changes in TF binding to the enhancer regions; importantly, even synonymous mutations exhibit such correlated changes. CRISPRi of selected EEs results in decreases in expression of host and/or neighboring genes. Finally, the authors investigate the evolutionary conservation of exonic enhancers.

Exonic enhancers are not a newly appreciated or otherwise novel phenomenon. They have been recognized for nearly 15 years, and at least one prior review has been dedicated to them. That review, however, is more than a decade old, and even though it (presumably) collected the sum of what was known about EEs at the time, it wasn't nearly as comprehensive or informative as this study. I'm not sure that there is a major question that might be asked regarding EEs that the authors haven't at least begun to address in this manuscript.

Comment 1

--Why should the peak of any given enhancer signature occur at the midpoint of an exon (Fig. 1E)? This would seem to have implications for how such sequences evolved.

We appreciate the reviewer's point and agree that the centered profile in Fig. 1e could give the impression of an artificial peak at the exon midpoint. In that figure, TF summit density was computed ± 1 kb from exon centers to accommodate exon length variability, but this normalization could visually bias signal toward the center.

To address this, we performed a new, non-centered metaprofile (Supplementary Fig. S6), aligning exons by their true 5' and 3' splice sites and rescaling them to a 0–100% coordinate. This approach avoids imposing symmetry around the midpoint. The resulting TF-summit distributions still showed a clear internal enrichment and consistent depletion near exon termini, observed across human, mouse, fly, and *Arabidopsis*. This may suggest a biological pattern: transcription factor binding tends to avoid splice junctions, potentially to minimize regulatory interference with splicing machinery.

While intriguing from an evolutionary standpoint, we prefer not to overinterpret this signal at this stage. We agree the internal positioning of TFs may reflect evolutionary pressures to separate regulatory activity from splicing elements. We intend to explore this further through future comparative and mechanistic analyses.

Comment 2

--Interesting that enhancer signatures peak just upstream of first coding exon (Figure S4), which would be expected, given the similarities between enhancers and promoters.

The reviewer is correct that the strong TF-ChIP enrichment immediately upstream of the first coding exon (left column of Supplementary Fig. S4) almost certainly reflects promoter architecture rather than enhancer activity. In vertebrates the first coding exon typically lies only ~100–300 bp downstream of the annotated transcription-start site, and the distance is even shorter in *Drosophila* and *Arabidopsis*. Any metaprofile centred on the first coding exon therefore captures promoter-proximal binding of general and sequence-specific transcription factors. We interpret this peak as a promoter hallmark that resembles an enhancer profile because promoters and enhancers share many chromatin and TF features. Although some promoters can function as enhancers (ePromoters), that phenomenon lies outside the scope of the present study.

To avoid confounding promoters with exonic enhancers, or including regions too close to transcription start sites (TSS) or transcription end sites (TES), our cEE catalogue applies stringent positional filters to coding exons (Methods). Specifically, we excluded all exons located within 1 kb of a TSS or TES in human and mouse, and within 100 bp of these landmarks in fly and *Arabidopsis*. This filtering step effectively removes the promoter-derived peak observed in Supplementary Fig. S4 and ensures that downstream cEE detection and analyses are not confounded by promoter-proximal signals.

Comment 3

--In Fig. 3C, EE19 appears to have more enhancer activity on the SV40 promoter than on its actual promoter (this is not a problem for the other EE tested).

EE19 robustly activates both its endogenous promoters (P1–P3) and the SV40 minimal promoter. The slightly higher induction observed with SV40 is expected: the SV40 *minimal* promoter has deliberately low basal output and is widely used in luciferase assays because it responds very strongly to heterologous enhancers, thereby maximizing dynamic range. Importantly, EE19's activity on SV40 is comparable to its activity on its strongest native promoter and does not alter our conclusion that EE19 functions as an enhancer.

To make this clear to readers we have:

- **Added a sentence to the Results (p. 5, lines 177–180):**
“Consistent with the promoter-independent behaviour expected of canonical enhancers, EE19 stimulates transcription equally well, indeed marginally better, when paired with the weak SV40 minimal promoter, which is well known to respond strongly to heterologous enhancer input.”
- **Expanded the Fig. 3C legend (p. 31 Line1077) to note the SV40 promoter's characteristics:**
“EE19 is slightly stronger with SV40 than with its native promoter, as expected for SV40's deliberately weak, enhancer-responsive design.

We hope this clarification fully addresses the reviewer's concern.

Comment 4

--"MobiDB27 analysis further supported this notion, showing that highly TF-bound EEs are frequently located in intrinsically disordered protein segments" – What on earth does that mean?

We thank the reviewer for pointing out that the original wording was opaque. What we meant is the following:

MobiDB analysis (intrinsic disorder) : We mapped every cEE onto the amino-acid sequence of its host protein and retrieved the per-residue disorder scores provided by MobiDB. cEEs that are bound by many different transcription factors (top quartile of TF occupancy in our ChIP-meta-analysis) fall disproportionately in regions predicted to be intrinsically disordered (IDRs).

AlphaFold analysis (structural flexibility) : The same regions also show lower AlphaFold pLDDT confidence scores (< 50), a hallmark of conformational flexibility/lack of fixed 3D structure. This independent metric therefore converges with the MobiDB prediction.

Biological interpretation : Intrinsically disordered segments tolerate more sequence variation without compromising protein folding. Locating cEEs in such segments provides an evolutionary “safe harbour”: the exon can acquire nucleotide-level motifs that recruit multiple TFs while the resulting amino-acid changes have minimal structural impact on the protein.

We have made changes in the paragraph (p. 10, Lines 378-384) with:

*“In contrast, mean pLDDT scores, AlphaFold's confidence metric for predicted 3D structure, declined steadily with increasing TF occupancy. **A second, orthogonal predictor of disorder (MobiDB) showed the same trend: the fraction of cEE residues classified as intrinsically disordered rose with TF occupancy. Both***

metrics indicate that cEEs that recruit many different TF are preferentially situated in intrinsically disordered, and therefore structurally flexible, protein regions. Because such regions tolerate amino-acid variation without disrupting protein folding, they provide an evolutionary “safe harbour” where nucleotide-level changes that create additional TF-binding motifs impose minimal structural cost on the protein.”

We hope this clarifies the initial confusion and clearly links the statistical observation enrichment of disorder to its functional implication: compatibility with enhancer evolution within coding sequences

Reviewer #4 (Remarks to the Author):

The manuscript by Mouren and colleagues focuses on exonic enhancers. Based on re-analysis of publicly available data, the authors show that many exons have an enhancer-like signature, incl. the presence of open chromatin, active histone modifications, and transcription factor binding. Further analysis of public data regarding human genetic variation and disease predisposition as well as cancer genomes identifies correlations between variants/mutations in exonic enhancers and disease predisposition/severity. Using reporter assays, incl. a high-throughput STARR-seq assay, the authors show that a subset of the exons with an enhancer signature can drive gene expression. Furthermore, the authors show that CRISPRi perturbations of exonic enhancers reduce gene expression. Finally, the authors perform evolutionary comparisons of exonic enhancer sequences.

Together, the authors provide a comprehensive overview of enhancer signatures in exons across species and their potential association with human disease, although the functional validation of these elements could have been more thorough. The manuscript is well written and the figures are clear and easy to follow. As discussed below, the main message of the manuscript is in my view not surprising per se, but I think the analyses presented are in principle of interest for the readership of Nature Communications.

Major comment 1

Although it is obvious that exonic DNA sequences have less flexibility to evolve into a potent enhancer, there is in principle no reason to expect that exonic DNA sequences could not function as an enhancer, especially since intronic enhancers are extremely widespread and exonic DNA and intronic DNA are not fundamentally different. For this reason, the findings of the manuscript are in my view not very surprising. I find the systematic assessment interesting nonetheless and am not at all opposed to the publication of this work, but I think the authors could consider framing their work more clearly in light of these comments.

We agree that, in principle, any genomic segment, including a coding exon, could acquire enhancer activity. Our work therefore does not present the notion “exons can be enhancers” as new. Instead, it closes the genome-wide quantitative gap that remained after those observations, and provides a **genome-wide, cross-species, and functional** portrait of how frequently, how strongly, and under what constraints such dual-function elements operate. To make this distinction clear we have (i) toned down any language that could be read as announcing a discovery, (ii) expanded the Introduction to place our work explicitly as a *genome-wide, comparative characterisation*, and (iii) added a short paragraph at the end of the result section that explain *why* the patterns we uncovered are unexpected and informative.

In a broader perspective, and also as an open discussion with the referee, we believe this study serves the community not only through its analyses but also by providing **direct access to genomic coordinates** of all identified candidate exonic enhancers (cEEs) across species, as well as **browser-ready visualizations**. While prior reports have described individual exonic enhancers, the regulatory genomics field has yet to fully integrate the idea that coding DNA can systematically encode cis-regulatory activity. In our view, this hesitation may reflect the lack of **robust, genome-wide datasets** and the absence of **standardized, reusable annotations**. By releasing a comprehensive and accessible catalogue, we aim to enable independent validation, reanalysis, and hypothesis generation. Whether future studies confirm, refine, or challenge our findings, we hope that this resource helps catalyze broader exploration of dual-function coding elements.

Manuscript changes

Introduction (p. 2, Lines 41-48, and Lines 49-55)

In the second paragraph (p. 2, Lines 41-48) we now cite the foundational exonic-enhancer literature (e.g. Lin *et al.* 2011; Ahituv 2016) and state that their existence is established. In the next following paragraph we position our study as a quantitative, comparative extension that reveals “how common, how conserved, and how functionally potent” these elements are.

Building on these pioneering locus-specific observations (cite Lin et al. 2011; Ahituv 2016), we undertook the first systematic, cross-species census of exons that simultaneously satisfy stringent enhancer criteria. Our analysis asks three quantitative questions that have not been addressed at genome scale: (i) How common are such dual-function exons? (ii) How potent and context-specific is their enhancer activity? (iii) What selective and mutational constraints accompany the coexistence of coding and regulatory information? By integrating >20,000 TF-binding profiles, multi-omic enhancer marks, high-throughput reporter assays, and comparative genomics, we provide a panoramic view of these candidate exonic enhancers (cEEs) across four phyla.

Results “Integrative paragraph” (end of Results) page 10, lines 386-400.

We added a short synthesis that focuses on what is new at scale :

We added a short subsection entitled “**What makes candidate exonic enhancers noteworthy?**”, highlighting four unexpected findings that emerged only from large-scale analysis:

Taken together, our analysis highlights several features that distinguish candidate exonic enhancers (cEEs) at genome scale. A minority of coding exons meet stringent enhancer criteria, yet cEEs are distributed across many genes. Despite harbouring fewer TF motifs than canonical intergenic enhancers (Fig. 2e; Supplementary Fig. 11), cEEs achieve comparable reporter activity (Fig. 4b), indicating efficient use of sequence under dual constraint. Population-genetic metrics indicate stronger contemporary purifying selection than in control exons (lower allele frequencies, higher synonymous/total ratio, lower pN/pS; Table S2; Supplementary Fig. S16,S21), while long-term patterns (higher dN/dS, positive DoS and FI > 1) point to episodic, compensatory amino-acid change. Finally, cEEs with the heaviest TF load preferentially fall in intrinsically disordered protein segments (Fig. 7f), suggesting that proteins accommodate embedded regulatory motifs in structurally flexible ‘safe harbours’.

Discussion (opening sentence, p. 11 line 402).

We removed the first sentence that might imply novelty of concept and now open with:

“Our genome-wide survey shows that exonic enhancers, although recognised at individual loci, represent a quantitatively substantial and functionally influential layer of regulation.”

Context paragraph in Discussion p.11 line 411-417.

We clarified why a genome-scale treatment is informative:

“Intronic and exonic DNA share the same chemical substrate, but coding exons carry additional informational load (reading frame, codon usage, and, for structured domains, side-chain constraints). Demonstrating that thousands of exons can accommodate dense TF-binding while preserving coding requirements, and quantifying the evolutionary and functional trade-offs that enable this, goes beyond documenting existence. Our analysis reveals how frequently the genome resolves this conflict, how strong the resulting enhancer outputs are, where within proteins such sequences reside, and which selective forces shape them.”

We believe these changes remove any impression of over-claiming conceptual novelty and clearly articulate the study’s contribution: a comprehensive, genome-wide, cross-species functional and evolutionary characterisation of enhancer-competent exons.

Major comment 2

In line with the previous point, one would expect that exonic enhancers would overall be less potent compared to enhancers in the non-coding genome as they have less flexibility to evolve into an optimal enhancer sequence.

This also seems evident from the STARR-seq comparisons (as the proportion of “Strong” exonic enhancers appears very small (5%) but the authors do not extensively discuss this.

We agree that coding sequence has less evolutionary freedom than non-coding DNA, so one might expect exonic enhancers to be weaker. However, in our K-562 STARR-seq screen the fraction of **Strong** elements is small for both classes and, if anything, slightly higher for cEEs: **61/2,068 (2.9%)** of exon-bounded inserts vs **21/1,184 (1.8%)** intergenic enhancers. Thus, potent activity is **not** suppressed in coding exons. Moreover, activity distributions overlap broadly and strong cEEs achieve reporter outputs comparable to canonical enhancers despite being shorter and carrying fewer TF motifs (Fig. 4b; Fig. 2e), consistent with efficient use of sequence under dual constraint.

Insertions into the manuscript

Results (p. 7 line 236-238):

“Among 2,068 exon-bounded inserts, 61 (2.9%) reached the ‘Strong’ activity bin, compared with 21 of 1,184 intergenic enhancers, indicating that high enhancer potency is not diminished within coding exons.”

Discussion p. 11 line 421-426 :

“Although ‘Strong’ elements constitute a small fraction in both libraries, as is typical for STARR-seq, exon enhancers reach strong reporter outputs at rates comparable to, or slightly exceeding, intergenic enhancers, despite tighter sequence constraints. Because we used exon-bounded fragments (no intronic sequence) and the same minimal promoter and thresholds across libraries, this comparison reflects intrinsic enhancer capacity rather than library design artifacts.”

Footnote for Figure 4 legend:

We are adding a short footnote in the Fig. 4 legend clarifying the Strong/Moderate/Weak thresholds so the reader immediately understands that the “Strong” class is intentionally stringent.

STARR-seq activity classes (“Strong”, “Moderate”, “Weak”) reflect log₂FC thresholds. The “Strong” class represents a deliberately stringent cutoff to capture only the highest enhancer outputs and constitutes a minority in all categories, consistent with prior STARR-seq studies.

In sum, the updates fix the numbers, address the critique, and demonstrate that coding constraint does not suppress strong enhancer output under matched assay conditions.

Major comment 3

Most of the analyses presented in the paper are correlative. The validations in the classic reporter assays and STARR-seq assay are nice, but it is well known that there is a poor correlation between reporter activity and in vivo activity. The best validation would be to assess the effects of synonymous point mutations in transcription factor binding sites in a native context (but I appreciated that this is a lot of work). The authors attempt to look at this in the context of cancer genomes, but given the large amount of mutations in this context, it is extremely difficult to assess the cause-consequence relationship. The CRISPRi experiments are in my view the only really informative functional assay in the manuscript. However, in these experiments, it is difficult to distinguish the effect of silencing the exonic enhancers from the effect that the recruited dCas9/KRAB complex may have on the transcribing polymerase. In this respect, it would be important to perform control experiments with gRNAs targeting exons that are not predicted to function as an enhancer. In addition, in these experiments it is not clear whether the authors are measuring nascent transcripts (with intronic primers) or stable transcripts (both are interesting from my point of view).

We acknowledge that reporter assays (luciferase, STARR-seq) are correlative and do not by themselves prove in vivo regulatory activity. However, to connect the two readouts, we overlaid the individually tested luciferase constructs onto the STARR-seq rank distribution (Fig. 4c) and observed that loci with >2× luciferase activity fall in the upper tail of the STARR-seq log₂FC distribution, indicating qualitative concordance between the two assays.

Manuscript insert (Results, STARR-seq paragraph, p. 7 line 239):

We observed 9 cEEs validated via luciferase assays (Fig. 3a) exhibited STARR-seq signals which are globally consistent with these activity thresholds (Fig. 4c). Top cEEs (FC>2x) fall in the upper tail of the STARR-seq rank distribution, indicating qualitative concordance between assays.

To our knowledge, our current STARR-seq assay represents the first genome-wide experimental validation of candidate exonic enhancers (cEEs), including a systematic evaluation of the effects of synonymous mutations motif-altering transcription factor binding sites. This is a first step in the genome wide characterization of EE.

Broad synonymous validation : A broader validation effort could involve a large-scale STARR-seq screen in multiple cell lines, and/or introducing systematic synonymous point mutations across all cEEs, allowing a comprehensive functional map of sequence variants affecting TF binding. While technically feasible, such an experiment would require synthesising and testing millions of constructs and remains beyond the scope of the current study.

Cancer analyses: Regarding our analysis of synonymous mutations in cancer genomes, we fully agree with the Reviewer that establishing direct cause–consequence relationships is challenging in this context. For this reason, our manuscript deliberately uses cautious language (e.g., “susceptible,” “potentially”) when discussing these associations, and we frame these findings as suggestive rather than conclusive.

Manuscript insertion (p. 8 line 308-312):

[We have addressed this point with referee’s #3 comments]

Collectively, these findings indicate that a subset of candidate exonic enhancers (cEEs) are somatically mutated in tumours and that such mutations are statistically associated with altered expression of nearby genes. Although causal links to tumour progression remain to be demonstrated experimentally, these observations highlight cEEs as genomically and transcriptionally responsive loci in cancer.

CRISPRi - Stable transcript : We apologize for the ambiguity; in all CRISPRi experiments we quantified stable transcripts. Expression changes were quantified at the **mature mRNA** level using exonic/ RT–qPCR primers.

Results insertion - CRISPRi subsection, p.9 line 332

*“We monitored the effects of cEE silencing on the **mature mRNA level** of the host (placing primers upstream and downstream of each cEE to capture potentially truncated transcripts) and neighbouring genes by RT–qPCR (Methods; Supplementary Table S5).”*

Methods insertion - CRISPRi targeting of EEs, p. 22 line 827-829

*“For CRISPRi readouts we measured **stable transcripts** by RT–qPCR with exonic primer pairs (not junction-spanning). Total RNA was DNase-treated prior to reverse transcription; primer sequences are listed in Supplementary Table S5.”*

Figure legend insertion (CRISPRi panel) p. 36 line 1158

“Bars show mature mRNA quantified by exonic RT–qPCR and compare measurements of gene expression [...]”

CRISPRi controls, intra-gene negative exons : To address the concern that dCas9/KRAB might non-specifically impede transcription (“roadblock”), we implemented matched intra-gene controls. For each gene in which we CRISPRi-targeted a cEE, we designed parallel gRNAs to a coding exon from the same gene that lacked obvious enhancer signatures (not a cCRE). One gRNA per exon was cloned, delivered and quantified with the same RT–qPCR readouts (Supplementary Fig. S28).

In this matched design, if KRAB/MecP2 recruitment caused a generic polymerase slowdown, both cEE-targeting and negative-exon gRNAs would comparably reduce host-gene output. Instead, non-cEE-directed guides did not

modify expression of host and/or predicted target genes (Supplementary Fig. S28). These intra-gene controls therefore argue for a locus-specific enhancer effect rather than a non-specific CRISPRi artefact.

Manuscript insertions:

Results (CRISPRi paragraph):

"In all four cases we observed a regulatory effect of the cEE inactivation, whereas matched gRNAs targeting non-cEE coding exons from the same genes (exons lacking obvious enhancer signatures) did not alter expression (Supplementary Fig. 28), supporting a sequence-specific enhancer effect."

Methods (CRISPRi design):

"For negative controls we used one sgRNAs to a same-gene coding exon lacking obvious enhancers signatures (such as TF binding, open chromatin, histone marks) from each of the tested cEE containing-gene (primers in Supplementary Table 8)."

Minor comments

1. Figure 1g: It is not clear what the fraction of exons and genes refers to here. What is the "total" to which these percentages refer?

We have revised the legend of Figure 1g to clarify that "% of Coding Genes" refers to the proportion of all annotated protein-coding genes that contain at least one cEE, and that "% of Coding Exons" refers to the proportion of all annotated protein-coding exons classified as cEEs. These changes improve clarity and ensure that the denominator is clearly defined for each metric.

Revised Figure 1g legend, p. 28 line 1030 :

*(g) Donut plots **showing** the proportion of coding exons (top) and coding genes (bottom) that contain at least one candidate exonic enhancer (cEE)s in each species. Percentages are calculated relative to the total number of annotated protein-coding exons (top) or genes (bottom) in the genome.*

2. What is the resolution of the promoter capture HI-C data? It seems unlikely that it is sufficient to distinguish interactions with the exonic enhancers and the surrounding regions. If that is indeed the case, conclusions about specific interactions should be toned down.

We agree with the reviewer that promoter capture Hi-C (PCHi-C) data are limited in resolution, particularly for pinpointing precise regulatory elements within short genomic intervals. The PCHi-C datasets from Laverré et al. (2022), which we used, report interaction bins ranging from 25 kb to 2 Mb. As such, while these data are informative at the domain level, they do not allow base-pair resolution mapping of enhancer-promoter contacts.

To address this, we have revised the relevant text in the manuscript to clarify that:

- The pHi-C data support **spatial proximity** between candidate exonic enhancers (cEEs) and candidate target promoters;
- However, they **cannot distinguish** whether the interaction arises from the exon itself or from adjacent regulatory sequences.

We have modified the following sentence to the Results section (p5) to make this limitation explicit:

Original:

"These analyses revealed physical contacts spanning tens of kilobases between each EE and its host gene promoter, as well as distal promoters, suggesting a long-range regulatory relationship."

Revised, p. 5 line 170-172:

“PChI-C mapping shows that each cEE lies in chromatin domains contacting its host-gene promoter and distal promoters, indicating long-range spatial proximity while, at the current 25kb–2Mb resolution, not ruling out contributions from nearby regulatory elements.”